# Open-o3 Video: Grounded Video Reasoning with Explicit Spatio-Temporal Evidence

## Abstract

Most video reasoning models only generate textual reasoning traces without indicating when and where key evidence appears. Recent models such as OpenAI-o3 have sparked wide interest in evidence-centered reasoning for images, yet extending this ability to videos is more challenging, as it requires joint temporal tracking and spatial localization across dynamic scenes. We introduce **Open-o3 Video**, a non-agent framework that integrates explicit spatio-temporal evidence into video reasoning, and carefully collect training data and design training strategies to address the aforementioned challenges. The model highlights key timestamps, objects, and bounding boxes alongside its answers, allowing reasoning to be grounded in concrete visual observations. To enable this functionality, we first curate and build two high-quality datasets, **STGR-CoT-30k** for SFT and **STGR-RL-36k** for RL, with carefully constructed temporal and spatial annotations, since most existing datasets offer either temporal spans for videos or spatial boxes on images, lacking unified spatio-temporal supervision and reasoning traces. Then, we adopt a cold-start reinforcement learning strategy with multiple specially designed rewards that jointly encourage answer accuracy, temporal alignment, and spatial precision. On V-STAR benchmark, **Open-o3 Video** achieves state-of-the-art performance, raising mAM by 14.4% and mLGM by 24.2% on the Qwen2.5-VL baseline. Consistent improvements are also observed on a broad range of video understanding benchmarks, such as VideoMME, WorldSense, VideoMMMU, LongVideo-Reason-eval and TVGBench. Beyond accuracy, the reasoning traces produced by Open-o3 Video also provide valuable signals for test-time scaling, enabling confidence-aware verification and improving answer reliability. The code and datasets will be made publicly available.

## 1 Introduction

Understanding complex video content is a long-standing goal for large multimodal models (Wang et al., 2025b; Team et al., 2025; Chen et al., 2024a; Zhang et al., 2024a; Ye et al., 2024; Zhang et al., 2023; 2024b; Wang et al., 2024), as videos encapsulate rich temporal dynamics and spatial interactions that far exceed the information in static images. While recent progress has advanced performance on tasks like action recognition and video question answering (Bai et al., 2025; Zhu et al., 2025; Zhang et al., 2024b; Li et al., 2024; Zhang et al., 2025a), building models that can perform reliable, fine-grained reasoning over long and cluttered scenes remains challenging.

Recent "thinking with images" attempts (OpenAI, 2025; Wang et al., 2025c;a; Zheng et al., 2025b) leverage explicit operations (such as cropping, zoom-in, and region selection) to interleave detailed *visual evidence* with language, achieving superior performance on fine-grained image comprehension. This success motivates extending a similar paradigm to the video domain.

However, this extension is difficult and non-trivial due to the requirement for *coherent localization across both time and space* precisely. The complexity of dynamic scenes, *e.g.*, replete with motion, occlusions, and camera changes, makes it incredibly challenging to pinpoint when and where events of interest occur. As a result, previous attempts to incorporate explicit reasoning in video have often been limited to *textual rationales* (Feng et al., 2025a; Wang et al., 2025d) or, coarse, *temporal-only* grounding (Li et al., 2025b; Wang et al., 2025f), failing to achieve the fine-grained spatio-temporal precision necessary for complex video reasoning. This gap is largely due to two interconnected

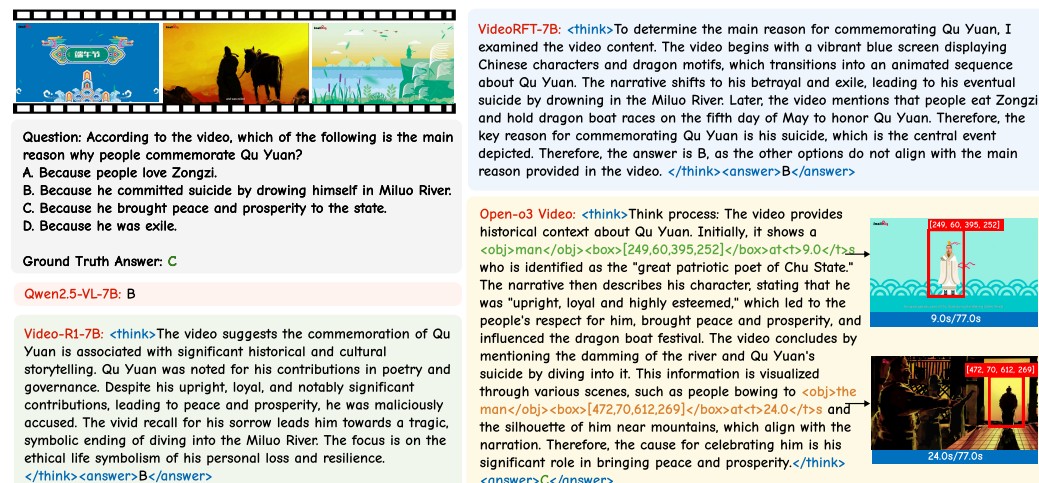

Figure 1: While prior video reasoning models (e.g., Video-R1 (Feng et al., 2025a), VideoRFT (Wang et al., 2025d)) only generate textual rationales, **Open-o3 Video** integrates explicit spatio-temporal grounding into the reasoning process. The model highlights key timestamps and object regions that directly support the answer, providing verifiable evidence for its prediction. More visualizations are provided in Appendix A.8.

obstacles: (1) the absence of *high-quality datasets* that provide joint spatio-temporal supervision for reasoning, and (2) the inherent difficulty of training a model to precisely localize objects in *time and space* simultaneously.

To address these challenges, we introduce **Open-o3 Video**, a framework that embeds *joint* spatio-temporal evidence directly into the reasoning process. Our first key contribution is the creation of a comprehensive training corpus designed to bridge this data gap. We have curated two datasets, STGR-CoT-30k and STGR-RL-36k, for supervised fine-tuning and reinforcement learning, respectively. These datasets integrate existing temporal-only and spatial-only grounding resources *with 5.9k newly annotated high-quality spatio-temporal samples*. Each instance contains a question-answer pair, timestamped key frames, localized bounding boxes, and *a chain of thought that explicitly links the visual evidence to the reasoning steps*.

Building on this dataset, our second major contribution is a two-stage training strategy with **adaptive temporal proximity** and **temporal gating** to stably and efficiently optimize the model's spatio-temporal reasoning capability. Although the model has acquired preliminary capabilities for generating structured, grounded chains of thought during the supervised fine-tuning stage, the subsequent reinforcement learning stage still cannot achieve stable training due to a critical *spatial collapse* issue. This is because spatial grounding rewards are usually conditioned on correctly identifying the timestamp. When temporal predictions are imprecise in the early stages, this leads to *near-zero spatial rewards*, stalling the learning process for localization. Therefore, we propose a novel *adaptive temporal proximity* technique, which relaxes the temporal requirement in early training to reduce reward sparsity, and gradually increases the precision demand over training time. This prevents premature saturation of the temporal reward and ensures that predicted timestamps keep approaching the ground truth, which is crucial for reliable spatial evaluation. In parallel, a complementary *temporal gating* mechanism computes spatial rewards only when temporal predictions are sufficiently accurate, preventing irrelevant objects from being rewarded and enforcing precise spatio-temporal alignment. Together, these mechanisms provide dense yet reliable feedback, forming a smoother learning curriculum that progressively strengthens both temporal accuracy and spatial grounding.

Through this combination of curated data and our training procedure, as shown in Figure 1, Open-o3 Video produces reasoning that is accurate, interpretable, and grounded in the visual evidence. We evaluate Open-o3 Video on the V-STAR benchmark and other video understanding tasks. On **V-STAR**, our model achieves state-of-the-art performance, surpassing GPT-4o and improving over Qwen2.5-VL by **+14.4%** mAM and **+24.2%** mLGM with a small amount of training data. Beyond V-STAR, Open-o3 Video also delivers consistent gains on VideoMME, WorldSense, VideoM-

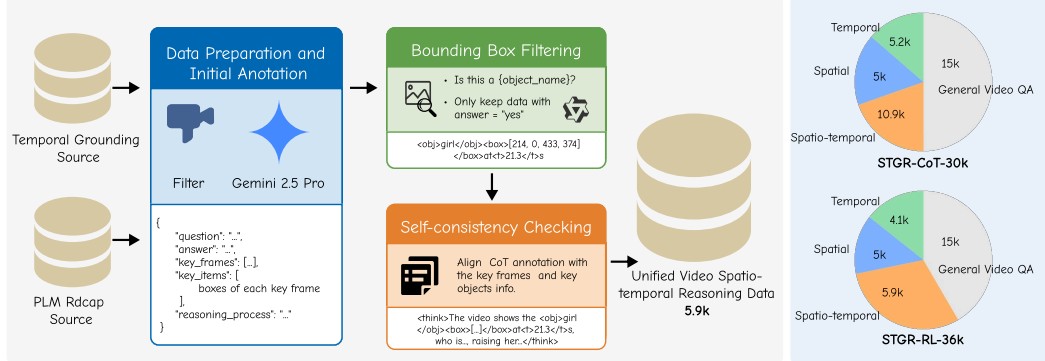

Figure 2: Overview of our data construction pipeline and dataset composition. **Left**: The annotation pipeline includes Gemini 2.5 Pro initial annotation, bounding box filtering, and self-consistency checking. **Right**: Distribution of data categories in STGR-CoT-30k (SFT) and STGR-RL-36k (RL), showing a balanced coverage across temporal, spatial, spatio-temporal, and general QA.

MMU, LongVideo-Reason-eval, and TVGBench, demonstrating advantages in long-video reasoning, perception-oriented tasks, and fine-grained temporal localization. In addition, the explicit evidence traces support evidence-aware test-time scaling, where confidence-aware voting surpasses majority voting (e.g., +1.2% on WorldSense and 1.0% on VideoMMMU), demonstrating that grounded evidence provides a reliable self-verification signal to improve inference accuracy.

## 2 RELATED WORKS

**Video Reasoning.** Recent advances in video reasoning (Feng et al., 2025b; Li et al., 2025b; Wang et al., 2025g;d; Zhang et al., 2025c; Xie et al., 2025; Chen et al., 2025b; Zhang et al., 2025b; Park et al., 2025; Dang et al., 2025) have largely been driven by reinforcement learning based post-training, which encourages models to move beyond direct question answering and exhibit step-by-step reasoning. Video-R1 (Feng et al., 2025b) shows that temporal-aware GRPO with curated reasoning data improves video understanding benchmarks, while VideoChat-R1 (Li et al., 2025b) extends to spatio-temporal perception tasks such as grounding and tracking without harming QA. Other variants, including Video-RTS (Wang et al., 2025g) and DeepVideo-R1 (Park et al., 2025), combine reinforcement learning with test-time scaling or difficulty-aware regularization to better exploit temporal information. These works demonstrate the potential of reinforcement-driven video reasoning, but still rely on text-only outputs without explicitly linking answers to visual evidence. In contrast, our approach generates spatio-temporal grounded evidence (timestamped frames and localized objects), enhancing perception, transparency, and verifiability.

**Temporal and Spatial Grounding in Video.** The problem of locating when and where relevant evidence appears in a video has attracted growing attention, leading to substantial progress in both temporal and spatial grounding (Wang et al., 2025f;e; Chen et al., 2025a; Guo et al., 2024; Ouyang et al., 2025; Li et al., 2025c;b). On the temporal side, Time-R1 (Wang et al., 2025f) introduces verifiable rewards for temporal grounding with strong generalization under limited supervision, while TVG-R1 (Chen et al., 2025a) improves robustness with curated cold-start and RL datasets. On the spatial side, SpaceR (Ouyang et al., 2025) leverages RL and a large corpus for object-centric grounding and geometric reasoning. Beyond these two sides, a number of approaches explore spatio-temporal localization. STCAT (Jin et al., 2022) and LRR (Bhattacharyya et al., 2023) improve the spatio-temporal grounding ability through model architectural optimization. EgoMask (Liang et al., 2025) enhances spatio-temporal localization for egocentric videos by fine-tuning models like Sa2Va (Yuan et al., 2025). LLaVA-ST (Li et al., 2025a) unify temporal and spatial grounding through positional embedding alignment and two-stream feature compression. However, these approaches do not combine such spatio-temporal localization ability with chain-of-thought reasoning. Aligning both timestamps and object regions within reasoning text, and further leveraging such grounded evidence to enhance video question answering, remain challenging. Our approach tackles both by explicitly linking boxes with temporal positions and integrating spatio-temporal evidence into reasoning, thereby strengthening perception and ensuring verifiability.

**Thinking with Images.** A growing line of research (OpenAI, 2025; Zheng et al., 2025b; Wang et al., 2025a;c; Fan et al., 2025) explores how multi-modal models improve reasoning by performing explicit visual operations such as cropping, zoom-in, and region selection, thereby producing intermediate evidence that is consumed within the reasoning chain. OpenAI-o3 (OpenAI, 2025) formalizes "thinking with images," while DeepEyes (Zheng et al., 2025b) shows end-to-end RL can incentivize image–tool reasoning, and TreeBench (Wang et al., 2025a) provides methodology for traceable, box-level evidence. These advances demonstrate the promise of evidence-centric visual reasoning but are largely image-centric. Extending to videos adds challenges in temporal consistency, motion, and fine-grained event alignment. VITAL (Zhang et al., 2025b) adapts the paradigm via an agent-based, tool-augmented RL pipeline, yielding gains but relying on external orchestration. In contrast, our single-model framework "thinks with frames," directly emitting timestamped crops and bounding boxes as evidence without complex tool pipelines.

## 3 STGR Data Construction

### 3.1 Data Source and Statistics

Building robust spatio-temporal reasoning models requires training signals that jointly supervise *when* and *where* evidence appears and how it is used in reasoning. Existing resources fall short in three ways: (i) temporal-only grounding datasets provide time spans but lack object regions; (ii) spatial or frame-level caption corpora offer boxes on isolated frames without timestamps; and (iii) most lack a chain of thought that *explicitly* ties objects and timestamps to the answer. These gaps make it impossible to learn coherent localization in dynamic scenes and to compute verifiable rewards for RL, since temporal and spatial supervision are not synchronized and reasoning traces are text-only.

To bridge this gap, we curate two complementary corpora: **STGR-CoT-30k** for supervised fine-tuning (SFT) and **STGR-RL-36k** for reinforcement learning (RL). Both combine existing temporal-only and spatial-only resources ***with 5.9k newly annotated, high-quality spatio-temporal samples*** produced by our pipeline (Sec. 3.2). Each new instance includes a question–answer pair, timestamped key frames, localized boxes, and a structured chain of thought that links visual evidence to reasoning steps. This design supplies synchronized temporal and spatial supervision for SFT to acquire grounded reasoning formats, and provides reliable, verifiable signals for RL to optimize alignment under complex video dynamics.

The SFT corpus consists of four components: (i) 4.1k temporal grounding CoT samples (TVG-Coldstart) (Chen et al., 2025a), (ii) 5k spatial grounding CoT samples (TreeVGR-SFT) (Wang et al., 2025a), (iii) 5.9k spatio-temporal samples curated by us, including 3.9k from temporal grounding datasets (video source: ActivityNet (Caba Heilbron et al., 2015), COIN (Tang et al., 2019), QueryD (Oncescu et al., 2021), QVHighlight (Lei et al., 2021), DiDeMo (Anne Hendricks et al., 2017)) and 2k from PLM-Rdcap (Cho et al., 2025), and (iv) 15k Video-R1-CoT samples (Feng et al., 2025b). The RL corpus further expands diversity: (i) 5.2k temporal grounding samples, including 2.3k from Time-R1 (Wang et al., 2025f) and 2.9k from TVG-RL (Chen et al., 2025a), (ii) 5k spatial grounding samples from VisCoT (Shao et al., 2024a), (iii) 10.9k spatio-temporal samples, comprising our 5.9k constructed data (via the pipeline) and an additional 5k filtered from VideoEspresso Han et al. (2025) with consistency checks, and (iv) 15k Video-R1 samples (Feng et al., 2025b).

Overall, as shown in Figure 2 (right), the SFT set covers 13.7% temporal, 16.7% spatial, 19.7% spatio-temporal, and 50.0% general QA data, while the RL set includes 14.4% temporal, 13.9% spatial, 30.3% spatio-temporal, and 41.7% QA data. This design ensures that both phases expose the model to diverse supervisory signals while emphasizing spatio-temporal reasoning as the central capability. More details about the training data are provided in the Appendix A.2.

### 3.2 Data Annotation Pipeline

Spatio-temporal reasoning requires chain-of-thought data that include both temporal spans and spatial grounding. We construct 5.9k such samples by combining temporal grounding datasets with PLM-Rdcap sources (Figure 2, left). The pipeline follows three stages below.

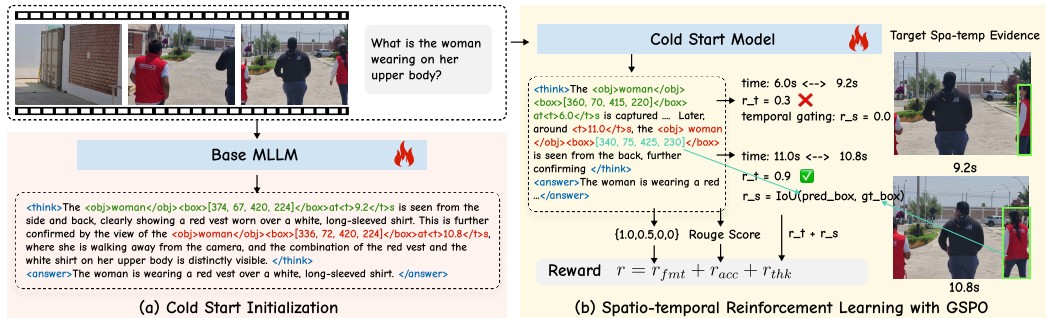

Figure 3: Overview of Open-o3 Video. We adopt a two-stage training paradigm: (a) cold-start initialization to learn structured, grounded outputs; (b) reinforcement learning with a composite reward that sharpens temporal alignment and spatial precision with adaptive temporal proximity and temporal gating.

**Data Preparation and Initial Annotation.** We begin by collecting two types of sources: temporal grounding datasets and PLM-Rdcap data that provide region-level dense captions. All videos are passed through Gemini 2.5 Pro (Comanici et al., 2025) API with carefully designed prompts (shown in the Appendix A.5) to generate structured annotations. Each annotation contains (i) a question-answer pair centered on a specific object or person, (ii) one to five key frames sampled from the annotated segment, (iii) bounding boxes for one to three salient objects in each key frame, and (iv) a reasoning process that must reference every object with explicit format: `<obj>object_name</obj><box>[x_min, y_min, x_max, y_max]</box>at<t>timestamp</t>s`.

**Bounding Box Filtering.** Initial annotations may contain noisy or incorrect boxes. We filter them with two rules: (i) boxes covering over 80% of the frame are removed as uninformative; (ii) each crop is verified by Qwen2.5-VL-7B (Bai et al., 2025) with the query "Is this a {object_name}?". Only samples answered "yes" are kept, ensuring object mentions match validated boxes.

**Self-consistency Checking and Quality Control.** Our consistency checking aims to align timestamps, bounding boxes, and the spatio-temporal reasoning chain. Since each annotated sample contains timestamps, object names, and a reasoning chain with temporal and spatial references, we first ensure that all boxes and frames mentioned in the reasoning text appear in the "key_frames" and "key_items" annotations; missing elements are removed to keep the annotations complete. We then evaluate the relevance between each visual evidence and the corresponding reasoning sentence. We parse the reasoning text, crop the referenced region, and ask Qwen2.5-VL whether the cropped image matches the sentence. If it is judged unrelated, we remove the sample. These consistency checks improve the quality of the reasoning data and support cold-start training for grounded spatio-temporal reasoning.

## 4 OPEN-O3 VIDEO

As shown in Figure 3, our training recipe involves two stages: a cold-start initialization phase followed by reinforcement learning to enhance spatio-temporal reasoning under carefully designed rewards with adaptive temporal proximity and temporal gating mechanisms.

### 4.1 COLD START INITIALIZATION

We initialize our model from Qwen2.5-VL-7B (Bai et al., 2025), and further fine-tune it on the constructed STGR-CoT-30k corpus. This stage yields checkpoints that equip the model with basic capabilities in spatio-temporal grounding and structured reasoning output. Such a cold-start stage is essential, as found in the experiment. It reduces reward sparsity, stabilizes optimization, and allows subsequent reinforcement learning to focus on fine-grained temporal and spatial alignment instead of relearning basic reasoning skills.

## 4.2 REINFORCEMENT LEARNING WITH GSPO

We adopt Group Sequence Policy Optimization (GSPO) (Zheng et al., 2025a) as our reinforcement learning algorithm. Compared with GRPO (Shao et al., 2024b), which operates at the token level, GSPO defines importance ratios and clipping at the sequence level, ensuring that optimization is aligned with sequence-level rewards. This eliminates high-variance token-wise corrections, stabilizes long-horizon training, and avoids collapse in chain-of-thought reasoning. Such stability is particularly important for video reasoning, where responses are longer, rewards combine accuracy, temporal, and spatial terms, and the training dynamics are more difficult to optimize. Our experiments further confirm that GSPO yields higher grounding accuracy and more stable training than GRPO (Section 5.2).

During training, given a video-question pair $x$, each generated response $y$ is evaluated with a scalar reward $r(x, y)$ that reflects both correctness and reasoning quality. This reward serves as the optimization signal in GSPO, and more details of the GSPO algorithm are provided in Appendix A.6.

## 4.3 REWARD DESIGN

For each query–completion pair $(x, y)$, the scalar reward is defined as

$$r(x, y) = r_{\mathrm{acc}}(x, y) + r_{\mathrm{thk}}(x, y) + r_{\mathrm{fmt}}(x, y), \tag{1}$$

which is group-normalized to obtain the advantage used by GSPO. Below we describe the three components.

**Accuracy reward $r_{\mathbf{acc}}$.** Since the training data span multiple tasks, we design task-specific accuracy rewards. For multiple-choice questions we check exact correctness; for free-form QA we follow previous works and compute ROUGE score; for spatial grounding we use visual IoU; and for temporal grounding we use temporal IoU:

$$r_{\mathrm{acc}}(x, y) = \begin{cases} 1 & \text{if task = MCQ and prediction matches ground truth,} \\ \mathrm{ROUGE}(y^{\mathrm{pred}}, y^{gt}) & \text{if task = Free-form QA,} \\ \mathrm{vIoU}(Box^{\mathrm{pred}}, Box^{\mathrm{gt}}) & \text{if task = Spatial grounding,} \\ \mathrm{tIoU}([s^{\mathrm{pred}}, e^{\mathrm{pred}}], [s^{\mathrm{gt}}, e^{\mathrm{gt}}]) & \text{if task = Temporal grounding.} \end{cases}$$

**Thinking reward $r_{\mathbf{thk}}$.** We define the thinking reward as the sum of temporal and spatial terms:

$$r_{\mathrm{thk}}(x, y) \; = \; r_{\mathrm{t}}(x, y) \; + \; r_{\mathrm{s}}(x, y). \tag{2}$$

*Temporal term with adaptive temporal proximity.* Let $M$ be the number of timestamps $\{t_m\}_{m=1}^M$ parsed from `<think>`. The temporal reward depends on the supervision type:

$$r_{\mathrm{t}}(x, y) \; = \; \begin{cases} \dfrac{1}{M} \displaystyle\sum_{m=1}^M \mathbf{1}\{\, s^{\mathrm{gt}} \le t_m \le e^{\mathrm{gt}} \,\}, & \text{interval supervision } [s^{\mathrm{gt}}, e^{\mathrm{gt}}], \\[2ex] \dfrac{1}{M} \displaystyle\sum_{m=1}^M \exp\!\Big( -\dfrac{\Delta t_m^2}{2\sigma^2} \Big), \quad \Delta t_m = \min_j |t_m - t_j^{\mathrm{gt}}|, & \text{point supervision } \{t_j^{\mathrm{gt}}\}, \\[2ex] 0, & \text{no timestamp evidence.} \end{cases} \tag{3}$$

A key difficulty is that spatial rewards depend on accurate temporal predictions: IoU can only be computed reliably when the timestamp is close to the ground truth. If the temporal constraint is too strict (i.e., $\sigma$ very small), the model receives little reward when its early temporal predictions are inaccurate, which slows down temporal learning and in turn prevents spatial grounding from being learned effectively. Conversely, if the constraint is always loose (i.e., $\sigma$ large), temporal rewards quickly saturate and stop driving predicted timestamps closer to the ground truth, which again undermines spatial reward reliability. To resolve this trade-off, we propose **adaptive temporal proximity**: $\sigma$ is large in early training to provide dense signals, and gradually decreases to enforce stricter alignment. This strategy ensures that the model first obtains stable gradients and later achieves precise timestamping, providing a solid foundation for spatial evaluation.

*Spatial term with temporal gating.* For each predicted timestamp $t_m$, let $j^\star(m) = \arg\min_j |t_m - t_j^{\text{gt}}|$ be the nearest annotated time. Let $\mathcal{B}_m$ be predicted boxes and $\mathcal{B}_{j^\star(m)}^{\text{gt}}$ ground-truth boxes on that frame. The spatial reward is

$$r_{\text{s}}(x,y) \;=\; \frac{1}{M}\sum_{m=1}^{M} \mathbf{1}\{\, |t_m - t_{j^\star(m)}^{\text{gt}}| \le \tau \,\} \cdot \max_{b \in \mathcal{B}_m,\, b^{\text{gt}} \in \mathcal{B}_{j^\star(m)}^{\text{gt}}} \text{IoU}(b, b^{\text{gt}}), \tag{4}$$

where $\tau$ is a temporal threshold. We further propose a **temporal gating** mechanism to guarantee the reliability of spatial supervision. Specifically, spatial rewards are only computed when temporal predictions are sufficiently close to the ground truth. This prevents rewarding salient but irrelevant objects at wrong timestamps, enforces spatio-temporal alignment, and ultimately improves both the interpretability and reliability of the reasoning process. Together, adaptive temporal proximity and temporal gating provide complementary solutions: the former supplies stable and progressive temporal supervision, while the latter ensures accurate and trustworthy spatial evaluation.

**Format reward $r_{\textbf{fmt}}$.** Strict usage of `<think>` and `<answer>` with correct `<obj>` `<box>` `<t>` gives 1.0. Having only `<think>` and `<answer>` yields 0.5. Otherwise, the reward is 0.0.

## 5 EXPERIMENTS

**Implementation Details.** We build upon the **Qwen2.5-VL-7B** model and train on 8 NVIDIA H100 GPUs. During training, we uniformly sample 16 frames from each video, where each frame has a resolution not exceeding $128 \times 28 \times 28$. If annotated key frames are available, they are inserted in addition to the uniformly sampled frames. To strengthen the model's perception of temporal information, we prepend each frame with its absolute timestamp. More implementation details are provided in Appendix A.1.

**Benchmarks.** We adopt V-STAR (Cheng et al., 2025) as the main benchmark, since it is specifically designed to measure spatio-temporal grounding in videos. Unlike conventional video QA datasets, V-STAR requires models to not only answer questions but also localize *when* and *where* the supporting evidence occurs. It introduces two structured reasoning chains ( "what–when–where" and "what–where–when") and composite metrics combining accuracy with temporal and spatial IoU, thereby enabling comprehensive evaluation of spatio-temporal reasoning. We further evaluate on broader video understanding benchmarks. VideoMME (Fu et al., 2025) and VideoMMMU (Hu et al., 2025) assess general video QA and multimodal comprehension across diverse domains, while WorldSense (Hong et al., 2025) emphasizes integrating multimodal signals with commonsense reasoning , and LongVideo-Reason-eval (Chen et al., 2025b) evaluates long-range reasoning on videos. In addition, TVGBench (Wang et al., 2025f) focuses on fine-grained temporal localization, STAR (Wu & Star, 2024) tests situated reasoning, and CameraBench (Lin et al., 2025) measures robustness under diverse camera motions.

### 5.1 MAIN RESULTS

**Results on V-STAR.** On the V-STAR benchmark, we compare our method with three groups of baselines: (i) closed-source commercial models such as GPT-4o (OpenAI, 2024) and Gemini-2-Flash (Team et al., 2024), which represent the current frontier of proprietary video LLMs. (ii) open-source general-purpose video understanding models, including Video-LLAMA3 (Zhang et al., 2025a), LLaVA-Video (Zhang et al., 2024b), VideoChat2 (Li et al., 2024), Oryx-1.5-7B (Liu et al., 2024), InternVL-2.5-8B (Chen et al., 2024b), and Qwen2.5-VL-7B (Bai et al., 2025). (iii) task-specialized approaches such as TRACE (Guo et al., 2024), designed for temporal video grounding, and Sa2VA (Yuan et al., 2025), optimized for fine-grained spatial grounding. As summarized in Table 1, our model consistently outperforms the baseline across different evaluation dimensions. In video question answering (*What*), our model achieves an accuracy of 61.03, representing a +27.6% point improvement over Qwen2.5-VL-7B. For temporal grounding (*When*), we report strong gains on both reasoning chains: Chain1 (*what–when–where*) improves by +9.1% points and Chain2 (*what–where–when*) by +10.2% points, showing robust performance regardless of the reasoning order. For spatial grounding (*Where*), our method surpasses the baseline by +8.4% points on Chain1 and +3.5% points on Chain2. Overall, compared with the Qwen2.5-VL baseline, our

Table 1: Performance on the **V-STAR** benchmark, which evaluates **spatio-temporal** reasoning across three dimensions. Chain1 denotes *what–when–where*, while Chain2 corresponds to *what–where–when*. mAM is the average of arithmetic mean, and mLGM is the average of modified logarithmic geometric mean, combining temporal and spatial alignment. [*] indicate we re-evaluate using the vLLM framework with 16 sampled frames. Bold numbers denote the best results, while underlined numbers indicate the second best.

| Model | What | When (Temporal IoU) | | Where (Visual IoU) | | Overall | |
|---|---|---|---|---|---|---|---|
| | Acc | Chain1 | Chain2 | Chain1 | Chain2 | mAM | mLGM |
| GPT-4o | 60.8 | 16.7 | 12.8 | 6.5 | 3.0 | 26.8 | 38.2 |
| Gemini-2-Flash | 53.0 | **24.5** | 23.8 | 4.6 | 2.2 | 26.9 | 35.6 |
| Video-LLaMA3 | 41.9 | 23.0 | 23.1 | 0.9 | 0.2 | 21.7 | 27.0 |
| LLaVA-Video | 49.5 | 10.5 | 12.2 | 1.9 | 1.3 | 20.8 | 27.3 |
| VideoChat2 | 36.2 | 13.7 | 12.5 | 2.5 | 1.0 | 17.0 | 20.3 |
| Oryx-1.5-7B | 20.5 | 13.5 | 14.8 | 10.1 | 3.5 | 15.1 | 13.8 |
| InternVL-2.5-8B | 44.2 | 8.7 | 7.8 | 0.7 | 0.1 | 17.6 | 24.9 |
| Qwen2.5-VL-7B[*] (base) | 33.5 | 15.4 | 13.8 | 17.0 | 2.5 | 19.3 | 22.4 |
| TRACE | 17.6 | 19.1 | 17.1 | 0.0 | 0.0 | 12.0 | 13.3 |
| Sa2VA-8B | 16.4 | 0.1 | 0.0 | **32.3** | **37.5** | 17.1 | 20.3 |
| Open-o3 Video (Ours) | **61.0** | **24.5** | **24.0** | 25.4 | 6.0 | **33.7** | **46.6** |
| Δ *vs.* Qwen2.5-VL-7B | ↑ 27.5 | ↑ 9.1 | ↑ 10.2 | ↑ 8.4 | ↑ 3.5 | ↑ 14.4 | ↑ 24.2 |

Table 2: Performance across different video understanding and temporal grounding benchmarks. "LRR" refers to LongVideo-Reason-eval Benchmark. More evaluation results (on STAR and CameraBench) are provided in Appendix A.3.

| Model | VideoMME | | WorldSense | | VideoMMMU | | LRR | TVGBench | Avg |
|---|---|---|---|---|---|---|---|---|---|
| | Overall | Long | Overall | Recognition | Overall | Perception | Acc | mIoU | |
| GPT-4o | 71.9 | - | 42.6 | - | 61.2 | 66.0 | - | - | - |
| VideoLLaMA3-7B | 60.6 | 48.7 | 37.3 | 38.1 | 46.5 | 59.7 | 59.8 | **22.2** | 45.3 |
| InternVL-2.5-8B | 62.3 | 51.2 | **39.6** | **38.5** | 42.4 | 57.0 | 62.0 | 6.3 | 42.5 |
| Qwen2.5-VL-7B (Base) | 62.4 | 50.8 | 36.1 | 33.7 | 51.2 | 64.7 | 59.3 | 16.3 | 45.1 |
| VideoRFT-7B | 59.8 | 50.7 | 38.2 | 36.6 | 51.1 | 66.0 | **69.4** | 14.3 | 46.6 |
| VideoR1-7B | 61.4 | 50.6 | 35.5 | 32.8 | **52.4** | 65.3 | 68.9 | 9.6 | 45.6 |
| Open-o3 Video (Ours) | **63.6** | **54.9** | 37.5 | 36.8 | 52.3 | **68.0** | **69.4** | 20.8 | **48.7** |
| Δ *vs.* Qwen2.5-VL-7B | ↑ 1.2 | ↑ 4.1 | ↑ 1.4 | ↑ 3.1 | ↑ 1.1 | ↑ 3.3 | ↑ 10.1 | ↑ 4.5 | ↑ 3.6 |

model improves performance by +14.4% mAM and +24.2% mLGM on V-STAR. It further surpasses proprietary models such as GPT-4o (OpenAI, 2024) and Gemini-2-Flash (Comanici et al., 2025) and achieves state-of-the-art performance. By extracting key frames and precise bounding boxes, Open-o3 Video brings o3-style, evidence-guided reasoning to videos, supplying more reliable and verifiable visual evidence during inference.

**Results on General Video Understanding and Temporal Grounding Benchmarks.** We further evaluate our method on a broad suite of video understanding benchmarks, comparing against three categories of baselines: (i) closed-source commercial models such as GPT-4o (OpenAI, 2024), (ii) open-source general-purpose video LLMs including Qwen2.5-VL-7B (Bai et al., 2025), Video-LLAMA3 (Zhang et al., 2025a), and InternVL-2.5-8B (Chen et al., 2024b), and (iii) recent reasoning-focused models such as VideoRFT-7B (Wang et al., 2025d) and VideoR1-7B (Feng et al., 2025b), which treat video understanding as text-only reasoning. In contrast, our method combines reasoning with explicit spatio-temporal grounding, enabling evidence-based inference. As shown in Table 2, Open-o3 Video achieves consistent improvements across all datasets. Across VideoMME, WorldSense, and VideoMMMU, our model shows consistent gains over Qwen2.5-VL-7B, with notable improvements on long videos (+4.1%) and perception-related tasks (+3.1% on WorldSense recognition and +3.3% on VideoMMMU perception), highlighting enhanced temporal reasoning and perceptual grounding. For long-range video reasoning, our model achieves 69.4% accuracy on the LongVideoReason-eval benchmark (LRR), and outperforms the baseline by +10.1%. Compared

Table 3: Ablation on Different training strategies.

| Setting | What | When (Temporal IoU) | | Where (Visual IoU) | | Overall | |
|---------|------|--------|--------|--------|--------|------|------|
| | Acc | Chain1 | Chain2 | Chain1 | Chain2 | mAM | mLGM |
| Baseline | 33.5 | 15.4 | 13.8 | 17.0 | 2.5 | 19.3 | 22.4 |
| Pure SFT | 53.0 | 19.6 | 17.2 | 23.3 | 4.6 | 28.5 | 37.1 |
| Pure RL (GSPO) | 56.4 | 21.6 | 20.7 | 23.7 | 3.7 | 30.4 | 40.7 |
| SFT+RL (GRPO) | 60.5 | 21.6 | 23.1 | 25.3 | 5.8 | 32.8 | 45.3 |
| SFT+RL (GSPO) | **61.0** | **24.5** | **24.0** | **25.4** | **6.0** | **33.7** | **46.6** |

<table>
<tr><td colspan="3">Table 4: Impact of two reward designs.</td></tr>
</table>

| Setting | mAM | mLGM |
|---------|-----|------|
| Open-o3 Video | **33.7** | **46.6** |
| w/o Ada. | 33.0 | 45.2 |
| w/o Gat. | 32.3 | 44.9 |

Table 5: Impact of spatio-temporal training data.

| Training data | mAM | mLGM |
|---------------|-----|------|
| w/o spatio-temporal data | 28.3 | 36.2 |
| + VideoEspresso | 31.1 | 43.6 |
| + Our annotated data | **33.7** | **46.6** |

with dedicated video reasoning methods, our model achieves comparable or even superior results, while providing more interpretable evidence in its reasoning process. On TVGBench, which directly measures temporal grounding, our model surpasses the baseline by a large margin (+4.5 mIoU), indicating significant gains in temporal localization. These results show that our approach **maintains the QA strength of general video LLMs** while enhancing the spatio-temporal grounding capability.

## 5.2 ABLATION AND ANALYSIS

**Training strategy: RL provides larger gains than SFT, while their combination yields the best results, with GSPO offering the most stable improvements.** As shown in Table 3, both SFT and RL substantially improve grounding over the base model. RL outperforms SFT (+2.1% mAM, +4.6% mLGM) by directly optimizing temporal and spatial alignment, while SFT ensures stable reasoning formats and basic grounding under supervision. Their combination is highly synergistic, reaching 33.7% mAM and 46.6% mLGM. Within this joint training, GSPO further surpasses GRPO (+0.9% mAM, +1.3% mLGM) by providing more stable rewards and better long-horizon temporal localization (+2.9% Chain1 tIoU).

**Reward design: Both adaptive temporal proximity and temporal gating are effective.** In the thinking reward, we introduce two mechanisms: adaptive temporal proximity (**Ada.**) and temporal gating (**Gat.**). To validate their effectiveness, we conduct ablation experiments on the V-STAR benchmark. Removing the proximity reward reduces performance by 0.7% mAM and 1.4% mLGM, showing that adaptive scaling helps the model better align predicted timestamps with annotated windows. Removing temporal gating causes larger drops of 1.4% mAM and 1.7% mLGM, confirming that gating is crucial for filtering irrelevant segments and preventing noisy spatial boxes. These results verify that our reward design effectively couples temporal and spatial grounding, leading to the strong performance.

**Training data: High-quality spatio-temporal annotations significantly boost grounding.** Without spatio-temporal (ST) supervision, the model exhibits substantially weaker performance, underscoring the necessity of Spatio-temporal annotations for effective grounding. Incorporating 9.6k filtered and rewritten *VideoEspresso* (Han et al., 2025) samples enables the model to perform basic spatio-temporal reasoning, leading to improvements of +2.8% mAM and +7.4% mLGM. Building upon this, we further construct 5.9k high-quality Spatio-temporal annotations through our dedicated pipeline (as illustrated in Figure 2), which bring a larger gain of +5.4% mAM and +10.4% mLGM. This shows the effectiveness of our pipeline and the critical role of high-quality spatio-temporal supervision.

**Test-time scaling with grounded evidence: Confidence-aware voting with Open-o3 Video outperforms naive majority voting.** Inspired by the scoring and adaptive voting mechanisms for video reasoning in CyberV (Meng et al., 2025), we introduce a confidence-aware voting scheme

that leverages grounded evidence to verify predictions at inference, as shown in Figure 7 in the appendix. Details, including scoring schemes, prompts, and results on WorldSense and VideoMMMU are provided in Appendix A.7.

## 6 CONCLUSION

We introduced **Open-o3 Video**, a unified framework for grounded video reasoning that generates explicit spatio-temporal evidence through timestamped frames and localized bounding boxes. With carefully curated high-quality training data, a two-stage strategy combining supervised fine-tuning and GSPO-based reinforcement learning, and novel thinking rewards incorporating adaptive temporal proximity and temporal gating, our method substantially improves answer accuracy, temporal alignment, and spatial grounding. Comprehensive experiments demonstrate that Open-o3 Video achieves state-of-the-art performance on the V-STAR benchmark, surpassing strong baselines including GPT-4o, while remaining broadly competitive across diverse video understanding tasks. In future work, we aim to further align reasoning chains across text, time, space, and audio modalities, and to extend our approach to more complex and longer video scenarios.

## ETHICS STATEMENT

All datasets used for evaluation in this work are publicly available benchmarks for video understanding. In addition, we construct a new dataset based on open-source data sources, which will be released to the community upon publication to ensure transparency and academic benefit. No private or personally identifiable information is involved, and all data usage strictly follows the intended research licenses. We also recognize potential risks such as biased annotations or unintended harmful outputs, and we emphasize that our method is intended solely for academic research.

## REPRODUCIBILITY STATEMENT

Comprehensive implementation details, including training procedures, hyperparameter configurations, and evaluation protocols, are provided in the main paper (Section 5) and Appendix A.1. Furthermore, upon acceptance of this paper, all source code, datasets, and trained model checkpoints will be made publicly available.

## LLM USAGE STATEMENT

Large language models (LLMs) are used solely to aid in polishing the writing of this paper, such as improving grammar, clarity, and readability. No LLMs are used for research ideation, experimental design, data analysis, or result generation. All technical contributions, experiments, and analyses are conducted entirely by the authors.

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

# A    APPENDIX

**Overview.** This appendix provides additional details and analyses to complement the main paper. Section A.1 gives more implementation details. Section A.2 describes training dataset preparation and ablations on the ratio of general VideoQA data. Section A.3 provides more experimental results on STAR benchmark and CameraBench. Section A.4 provides analysis of inference frame rate. Section A.5 presents the prompts used for data annotation with Gemini. Section A.6 provides the full mathematical formulation of GSPO algorithm. Section A.7 details the confidence-aware test-time scaling procedure and reports additional results. Section A.8 provides further qualitative visualizations of spatio-temporal reasoning. Finally, Section A.9 discusses limitations of our current framework and directions for future work.

## A.1    MORE IMPLEMENTATION DETAILS.

The training process of Open-o3 Video consists of two stages. In the cold-start stage, we train on the STGR-CoT-30k dataset for one epoch with a learning rate of $1 \times 10^{-6}$. In the GSPO stage, we further train on the STGR-RL-36k dataset for one epoch, also with a learning rate of $1 \times 10^{-6}$. For the thinking reward, the standard deviation parameter $\sigma$ is annealed from 4 to 1 and then kept constant. The gating mechanism employs a temporal threshold $\tau$ of 3s. At test time, we employ the vLLM framework, requiring the model to first produce a spatio-temporal grounded reasoning process, followed by the final answer.

## A.2    MORE DETAILS AND ABLATION ON TRAINING DATA.

**Data Preparation.** Beyond reporting corpus sizes, we describe here the sampling and filtering strategy applied to each source. For temporal grounding data, we adopt strict constraints to ensure annotation quality and manageable reasoning length. Specifically, for TVG-Coldstart, we retain only samples with chain-of-thought length under 6,000 characters and with ground-truth spans covering less than 70% of the total video duration. The same filtering is applied to Time-R1, resulting in 2.3k samples. For additional temporal grounding video sources (ActivityNet, COIN, QueryD, QVHighlight, and DiDeMo), we keep videos of duration between 10 seconds and 3 minutes, further discarding those where the annotated action lasts more than 50% of the video; TVG-RL is filtered with the same rules, and 2.9k samples are randomly selected. For spatial grounding data, we randomly sample 5k instances from both TreeVGR-SFT and VisCoT. For general video QA data, 15k Video-R1 samples are randomly drawn without additional filtering. For PLM-based video dense captioning data (PLM-Rdcap), we initially sample 3k videos for annotation, from which 2k remain after filtering for quality and consistency. This careful selection yields a high-quality dataset that balances temporal, spatial, and general reasoning tasks. The resulting dataset provides diverse yet clean supervision signals, making it particularly suitable for training and evaluating spatio-temporal reasoning models.

**Ablation on Different Ratios of General VideoQA Data.** To enhance the model's grounding ability, we emphasize temporal and spatial grounding data during training. However, excessive focus on grounding may weaken the model's original strength in general VideoQA. Thus, an important design choice is how much general VideoQA data to include in the STGR dataset. We compare different ratios and evaluate performance on both grounding-oriented (VSTAR) and QA-oriented (VideoMME) benchmarks. As shown in Table 6, adding 15k VideoQA samples significantly improves QA accuracy without harming grounding performance. In contrast, adding 30k yields no further QA gain while slightly reducing grounding accuracy. Therefore, we adopt 15k VideoQA samples as a balanced choice, offering strong QA capability while preserving grounding ability, and maintaining training efficiency.

## A.3    MORE EVALUATION RESULTS

As shown in Table 7, Open-o3 Video performs better than the base model on both STAR and CameraBench. On STAR, it improves accuracy by +3.2%, showing that Open-o3 Video can better handle situated reaoning tasks when involving spatio-temporal cues. We also evaluate our model on the CameraBench VQA task and compare it with the baseline model as well as models trained without adaptive temporal proximity or without temporal gating. We find that our model performs better

Table 6: Impact of different amounts of general VideoQA data. 15k achieves the best balance between grounding and general QA performance.

| VideoQA Data | VSTAR (mAM) | VideoMME (Acc) |
|---|---|---|
| w/o Video-R1 data | 33.4 | 60.7 |
| +5k | 33.0 | 63.2 |
| +15k | **33.7** | **63.6** |
| +30k | 31.7 | **63.6** |

Table 7: Performance on STAR and CameraBench.

| Models | STAR | CameraBench VQA | | | |
|---|---|---|---|---|---|
| | Overall | Overall | Confusable Motion | Motion and Steadiness | Motion Speed |
| Qwen2.5-VL-7B | 67.3 | 57.5 | 49.3 | 56.7 | 69.0 |
| Open-o3 Video | **70.5** | **58.8** | **51.3** | **57.6** | **69.3** |
| w/o Ada. | 70.1 | 58.5 | 50.0 | 56.7 | 68.7 |
| w/o Gat. | 69.6 | 57.8 | 50.3 | 55.9 | 67.0 |

than the baseline and shows gains in challenging motion settings, such as confusable motion, motion and steadiness, and different motion speeds. It also outperforms the variants without Ada. or without Gat. These results indicate that both the model and the training techniques remain stable under camera motions that differ from the training data distribution.

### A.4 Ablation Studies on Inference Frame Rate

We analyze the effect of inference frame rate on long-video understanding using the LongVideo-Reason-eval benchmark, as shown in Table 8. For the comparison between high and low frame rates, we find that higher frame rates (64 frames) give some improvement, but even with only 16 frames, our model performs well and surpasses the baseline and other reasoning models. And the gain of increasing more frames is small. For variable frame rates, we follow AKS (Tang et al., 2025) and apply the key frames selection strategy. This strategy reaches 70.1% accuracy, showing that key frames sampling can offer a small improvement over uniform sampling when involving spatio-temporal reasoning.

### A.5 Prompt for Data Annotation.

To obtain high-quality spatio-temporal annotations, we design structured prompts for the Gemini 2.5 Pro API, separately tailored to the two data sources described in Section 3: PLM-Rdcap data and temporal grounding datasets. The goal of these prompts is to guide the model to produce question-answer pairs, key frame selection, bounding boxes, and reasoning chains in a consistent JSON format.

For PLM-Rdcap, as shown in Figure 4, the input is the dense video captions and total frame count, and the output is a JSON with *question*, *answer*, *key_frames*, and *reasoning_process*. Since only frame indices are given, we post-process them into timestamps and align reasoning mentions with annotated object names and boxes.

For temporal grounding datasets, as shown in Figure 5, the input includes the annotated segment, video duration, and segment descriptions, and the output JSON contains the *question*, *answer*, *key_frames* with timestamps, objects and boxes, and the spatio-temporal grounded *reasoning_process*.

We further apply strict filtering and consistency checks, retaining only annotations with validated boxes, aligned timestamps, and coherent reasoning. This ensures a high-quality dataset with reliable spatio-temporal evidence, essential for robust training and evaluation.

Table 8: Ablation on inference frame rate on LongVideo-Reason-eval.

| Models | Qwen2.5-VL | Video-R1 | | VideoRFT | | Open-o3 Video | | |
|---|---|---|---|---|---|---|---|---|
| Number of Frames | 64 | 64 | 16 | 64 | 16 | 64 | 64 (+AKS) | 16 |
| LongVideo-Reason-eval | 59.3 | 68.9 | 67.3 | 69.4 | 68.0 | 69.4 | 70.1 | 69.2 |

**Prompt for Gemini 2.5 Pro (PLM-Rdcap)**

The video contains a total of {item['total_frames']} frames, with the following dense captions information:
{str(dcap)}
Please complete the following tasks based on the video and caption information:
1. Generate a question-answer pair. Since the dense caption is centered on a specific object or person, the question should also focus on this object or person. You can consider aspects such as its color, clothing, actions, and so on.
2. Output key_frames, which should be the critical frames needed to answer the question. The key_frames must be a list of integer values and fall within the frame range mentioned in the dense caption. (at least one and at most five).
3. Generate a reasoning process:
  - Reasoning must use visual evidence grounded in the video.
  - When referencing the target object or person, you MUST use the following strict format: <obj>object_name</obj>at<t>Frame frame_num</t>
  - The reasoning must not exceed 200 words.
  - The frame number must be in key_frames. The mentioned frame numbers and the visual content of those frames must match consistently.
  - All object names must be identical.
  - Every time you mention the object name (<obj>), you must use the format `<obj>object_name</obj>at<t>Frame frame_num</t>` to specify the corresponding frame.
  - In the reasoning process, except for the text between <t> </t>, the words "frames", "frame" and similar terms MUST not appear.

You must strictly follow the following JSON format (with no additional text outside the JSON):
{{
    "question": "…",
    "answer": "…",
    "key_frames": […],
    "reasoning_process": "…"
}}

Figure 4: Annotation Prompt for PLM-Video-Human Region Dense Temporal Captioning Data source.

## A.6 DETAILS OF GSPO TRAINING

For completeness, we provide the full formulation of Group Sequence Policy Optimization (GSPO) (Zheng et al., 2025a), which is used in our reinforcement learning stage.

Given a query $x$, the model generates a group of $G$ candidate responses $\{y_i\}_{i=1}^{G}$ sampled from the old policy $\pi_{\theta_{\text{old}}}(\cdot|x)$. Each response is scored by a reward function $r(x, y_i)$, and its normalized advantage is computed as

$$\hat{A}_i = \frac{r(x, y_i) - \text{mean}(\{r(x, y_j)\}_{j=1}^{G})}{\text{std}(\{r(x, y_j)\}_{j=1}^{G})}. \quad (5)$$

The importance ratio is defined at the sequence level as

$$s_i(\theta) = \left( \frac{\pi_\theta(y_i|x)}{\pi_{\theta_{\text{old}}}(y_i|x)} \right)^{\frac{1}{|y_i|}} = \exp\left( \frac{1}{|y_i|} \sum_{t=1}^{|y_i|} \log \frac{\pi_\theta(y_{i,t}|x, y_{i,<t})}{\pi_{\theta_{\text{old}}}(y_{i,t}|x, y_{i,<t})} \right), \quad (6)$$

where $|y_i|$ denotes the response length.

The GSPO objective is then

$$J_{\text{GSPO}}(\theta) = \mathbb{E}_{x, \{y_i\} \sim \pi_{\theta_{\text{old}}}} \left[ \frac{1}{G} \sum_{i=1}^{G} \min\left( s_i(\theta)\hat{A}_i, \ \text{clip}(s_i(\theta), 1 - \epsilon, 1 + \epsilon)\, \hat{A}_i \right) \right], \quad (7)$$

```
Prompt for Gemini 2.5 Pro (Temporal Grounding)

The video has a duration of {item['duration']} seconds. The temporal grounding annotation for the video is as follows:
Description: {item['conversations']}
Annotated time segment: {str(item['gt_segment'])}

Based on this annotation, please complete the following tasks:

1. Construct a question-answer pair in open-ended Q&A format.
   - The question should be adapted from the temporal grounding description.
   - The question should focus on a specific object or person, rather than their action.
   - The answer should be concise and not exceed 30 words.
   - Do NOT mention timestamps or annotated time segments in the question.

2. Select at least ONE and at most FIVE keyframes.
   - Each timestamp MUST be within the annotated time segment and be written as a float rounded to exactly one
decimal place.

3. For each keyframe, include at least ONE and at most THREE detected objects.
   - Each bounding box coordinates are normalized floats (rounded to exactly two decimal place) in the format
[x_min, y_min, x_max, y_max].

4. Generate the reasoning process for answering the question:
   - Reasoning must use visual evidence grounded in the video.
   - When referencing any object, person, or visual element, you MUST use the following strict format:
     <obj>object_name</obj><box>[x_min, y_min, x_max, y_max]</box>at<t>time_in_seconds</t>s
   - Both the `time_in_seconds` and the box coordinates MUST be consistent with the info in the key frames.
   - The reasoning must not exceed 200 words.

You must strictly follow the following JSON format (with no additional text outside the JSON):

{{
    "question": "…",
    "answer": "…",
    "key_frames": [
        {{
            "timestamp": time in second,
            "items": {{
                "object1_name": box1,
                "object2_name": box2,
            }}
        }},
        {{
            "timestamp": time in second,
            …
        }}
    ],
    "reasoning_process": "…"
}}
```

Figure 5: Annotation Prompt for Temporal Grounding Data Source.

with $\epsilon$ controlling the clipping range.

Unlike GRPO, which clips per-token updates, GSPO clips entire responses, thereby aligning reward assignment with optimization granularity. In practice, this leads to more stable gradients and better performance on long chain-of-thought reasoning tasks.

## A.7 MORE DETAILS ABOUT TEST TIME SCALING.

To further enhance robustness at inference, we adopt a **confidence-aware test-time scaling** procedure, as shown in Figure 7. Given a video question, the model first generates $N$ independent responses in parallel (In our experiments, $N = 8$, with temperature set to 1.0). Each response contains spatio-temporal grounding annotations in the format

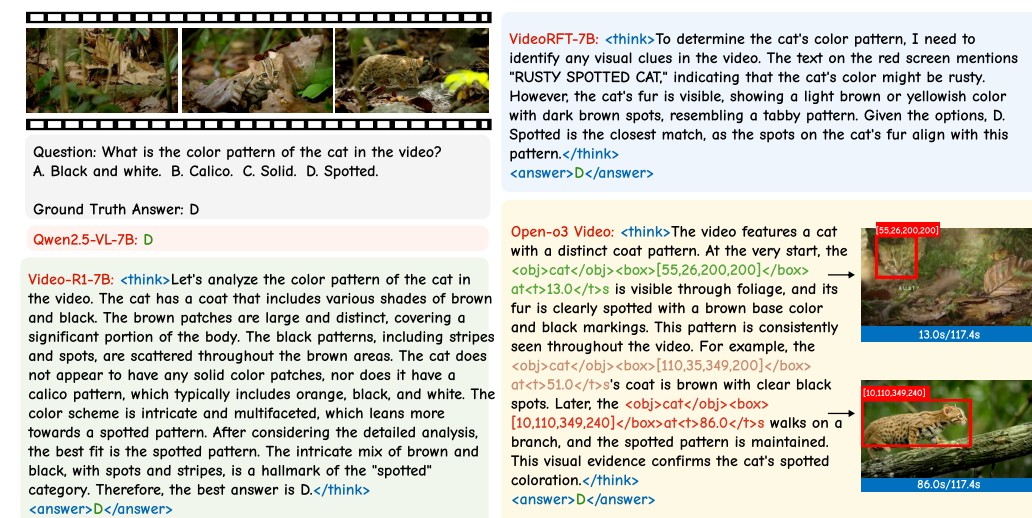

Figure 6: **Visualization.** On simple appearance perception tasks, both our model and related baselines can provide correct answers; however, our approach additionally offers explicit spatio-temporal evidence.

Table 9: Test-time scaling results on WorldSense and VideoMMMU, showing that the confidence-aware voting (N=8) with grounded evidence consistently outperforms base model (N=1) and naive majority voting (N=8).

| Setting | WorldSense | VideoMMMU |
|---|---|---|
| Base | 37.5 | 52.3 |
| Majority Voting | 37.3 | 53.1 |
| Confidence-aware Voting | **38.5** | **54.1** |

`<obj>...</obj><box>...</box>at<t>...</t>`s, from which we extract the predicted bounding boxes. The corresponding regions are then cropped from the original video frames and paired with the question to form a new input. This input is passed back into the model to obtain a confidence score $s \in \{0, 1, 2\}$, where:

- $s = 2$: the cropped evidence is highly supportive for answering the question,

- $s = 1$: the evidence may be partially useful,

- $s = 0$: the evidence is irrelevant.

Each initial response is assigned a confidence-weighted score by averaging its evidence scores across all mentioned objects. The final prediction is selected via weighted voting over the $N$ responses. This process effectively filters out hallucinated reasoning traces and highlights consistent evidence across responses.

As reported in Table 9, confidence-aware voting consistently improves over *naive majority voting*, achieving +1.0 on WorldSense and +1.0 on VideoMMMU. This demonstrates that our o3-style spatio-temporal evidence not only enhances grounding, but also provides a natural mechanism for scalable inference and self-correction at test time.

### A.8    MORE VISUALIZATIONS.

As shown in Figure 6,8,9, we provide additional qualitative examples to illustrate the spatio-temporal reasoning ability of Open-o3 Video. These visualizations demonstrate that our model can obtain spatio-temporal evidence and achieve better results.

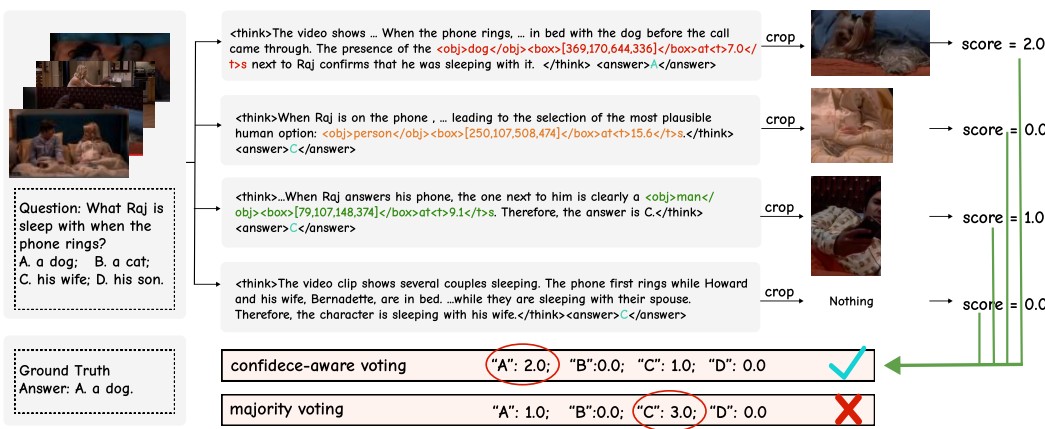

Figure 7: Illustration of our **confidence-aware test-time scaling**. The model generates multiple responses with spatio-temporal traces, from which visual regions are cropped and scored for evidence relevance ($s \in \{0, 1, 2\}$). Final predictions are obtained by confidence-weighted voting. Unlike naive majority voting that is misled by spurious patterns (predicting "C"), our method highlights consistent supportive evidence and correctly predicts "A", improving robustness at inference.

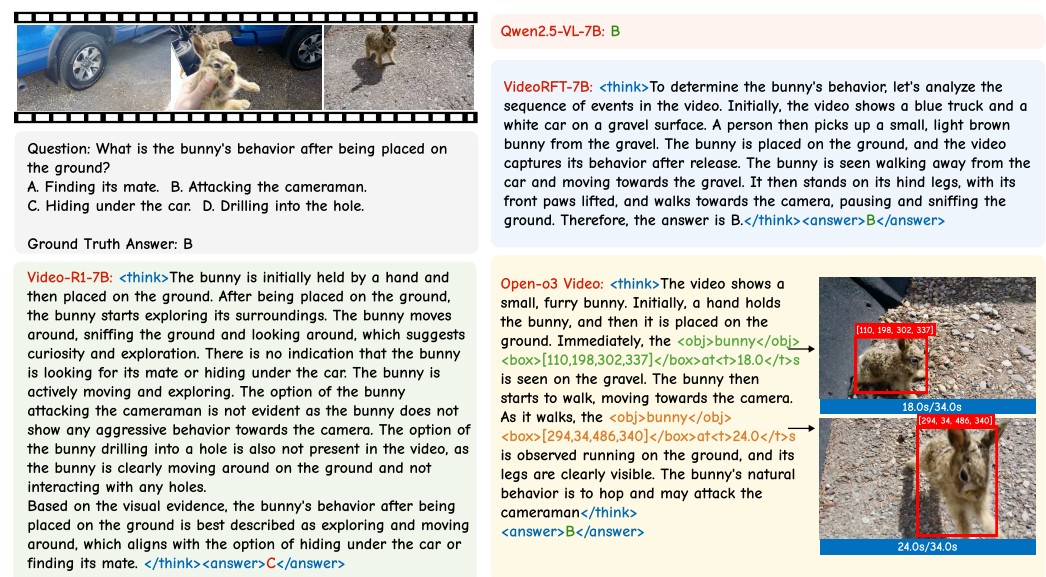

Figure 8: **Visualization.** For action recognition, our model precisely localizes both the time and location of the action, achieving superior performance compared to Video-R1.

## A.9    LIMITATIONS AND FUTURE WORK.

While our framework demonstrates strong performance, several limitations remain. First, handling longer videos with complex scenes and smaller objects is still challenging, as high-quality spatio-temporal data for such cases is still relatively scarce. Second, reasoning-intensive queries that require multi-step inference beyond direct grounding remain difficult to fully address. Finally, our current design does not integrate audio or speech information, which often carries crucial cues for understanding video content. Future work will focus on extending the approach to longer and more

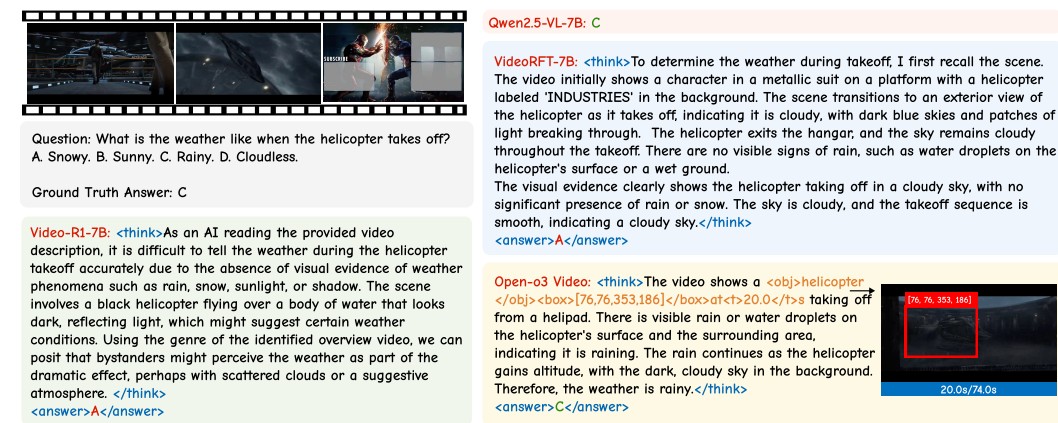

Figure 9: **Visualization.** In weather reasoning tasks, our model identifies more effective supporting evidence, whereas related video reasoning models perform poorly.

complex videos, enriching supervision for fine-grained object grounding, and unifying multimodal signals including speech to further enhance logical reasoning.

