# OpenReview forum: "Open-o3 Video: Grounded Video Reasoning with Explicit Spatio-Temporal Evidence"
_ICLR.cc/2026/Conference — Submitted to ICLR 2026_

### Official Review · Reviewer_BpJt · 2025-10-16

**Soundness:** 3
**Presentation:** 2
**Contribution:** 3
**Rating:** 4
**Confidence:** 4

**Summary:**

This paper presents the Open-o3 Video framework, designed for grounded video reasoning with explicit spatio-temporal evidence. It tackles the challenges of temporal and spatial grounding in dynamic video scenes, introducing two new datasets—STGR-CoT-30k for supervised fine-tuning (SFT) and STGR-RL-36k for reinforcement learning (RL). The model integrates spatial and temporal information into the reasoning process, achieving state-of-the-art performance across various video reasoning benchmarks.

**Strengths:**

1. The integration of spatio-temporal evidence directly into the reasoning process is a major innovation, addressing a long-standing challenge in video understanding. The use of both spatial and temporal alignment as core components of the training strategy offers a clear advancement over previous methods that focused on either dimension.
2. The introduction of the STGR-CoT-30k and STGR-RL-36k datasets provides the research community with a valuable resource for training spatio-temporal models, filling a gap that has hindered the progress of grounded video reasoning.
3. Open-o3 Video achieves state-of-the-art results across multiple benchmarks, demonstrating clear performance improvements over previous models like Qwen2.5-VL and GPT-4o. This positions the model as a strong candidate for real-world applications where interpretability and robustness are key.

**Weaknesses:**

1. During the initial training phase, the model faces a cold start issue, especially in reinforcement learning (RL) where inaccurate temporal predictions result in near-zero spatial rewards, preventing effective learning of spatial localization in the early stages. Although adaptive temporal proximity is introduced to alleviate this, it may still impact model stability, especially in complex scenes or noisy data, limiting real-time applications.
2. Open-o3 Video relies on high-quality spatio-temporal annotation data, which is still scarce, particularly for long videos and complex scenes. The lack of annotations for small objects and long-duration scenes may affect the model's performance and generalization ability.
3. When handling longer videos and complex scenes, the model still faces issues with spatial localization and temporal alignment, especially in small object or fast-moving scenarios. Despite enhancements in the reward mechanism, the model's performance in these challenging scenarios remains limited and requires further optimization.

**Questions:**

1. How do the adaptive temporal proximity and temporal gating mechanisms affect the model’s ability to work with videos that are different from the dataset (e.g., videos with unusual scenes or unexpected camera movements)?
2. Can the model perform better if multi-modal data, like audio, is added? This could help the model with videos that have important non-visual information.
3. The model works well with short and medium-length videos. How will it handle very long videos (e.g., full-length films) in real-world situations, where memory and computational limits are important?

---

> ### Author Response · Authors · 2025-11-23
> **Official Comment by Authors (Part1)**
>
> We sincerely thank the reviewer BpJt for the detailed and thoughtful comments. We appreciate the recognition of our main contributions, including the integration of spatio-temporal evidence into the reasoning process, the annotated datasets, and the performance on multiple benchmarks. Your feedback is very valuable, and we respond to each weakness and question below.
>
> **W1 and Q1: About the cold-start issues and generalization to unusual videos.**
>
> The concern is whether adaptive temporal proximity and temporal gating might reduce model stability when videos differ from the training distribution, such as unusual scenes or unexpected camera motions. To evaluate this, we use CameraBench [1], a recent benchmark designed to test robustness under non-typical camera motions, including unusual viewpoints, confusable motion patterns, camera movements, and varying motion speeds.
>
> We evaluate Open-o3 Video on the **CameraBench ​VQA task**​, and compare it with the baseline model as well as models trained **without adaptive temporal proximity or without temporal gating.** The results are shown below:
>
> |                 | Overall Acc    | Confusable Motion Acc | Motion and Steadiness Acc | Motion Speed Acc |
> | --------------- | ---------------- | ----------------------- | --------------------------- | ------------------ |
> | Qwen2.5-VL-7B | 57.5           | 49.3                  | 56.7                      | 69.0             |
> | Open-o3 Video | **58.8** | **51.3**        | **57.6**            | **69.3**   |
> | w/o Gat.      | 57.8           | 50.3                  | 55.9                      | 67.0             |
> | w/o Ada.      | 58.5           | 50.0                  | 56.7                      | 68.7             |
>
> *Table: Results on CameraBench VQA, comparing Open-o3 Video with the baseline and ablations without temporal gating (Gat.) or adaptive temporal proximity (Ada.)*
>
> From these results, we find that:
>
> (1) On videos outside the training distribution and under different types of camera motion, Open-o3 Video performs better than the baseline model.
>
> (2) Models trained without adaptive temporal proximity (Ada.) or without temporal gating  (Gat.) show lower performance, indicating the value of these two mechanisms.
>
> (3) In several representative subsets, such as confusable motion, motion and steadiness, and different motion speeds, our model improves by about **1–2%** over models without these techniques.
>
>
> These results show that the model remains stable under different and rapidly changing camera motions, and  technical mechanisms can help improve its robustness. We have appended these results in the updated draft.
>
>
> **W2: About data scarcity and generalization to long or complex scenes.**
>
> We thank the reviewer for pointing out this limitation. We acknowledge that spatio-temporal annotation data are still relatively scarce, and there is significant room for improvement, especially for long videos, small objects, and more complex scenes. This is also highlighted in the Limitations and Future Work sections of our paper (Appendix A.9) Constructing datasets with richer annotations for small objects and long-duration scenes will be a priority in our future work.
>
> At the same time, even with a limited amount of training data and without very long videos, the model still shows **certain generalization ability to long video scenarios.** On **VideoMME-Long** and ​**LongVideo-Reason-Eval** [2]​, the model improves over the Qwen2.5-VL baseline by +4.1% and ​+10.1%​, respectively, and reaches or exceeds the performance of other video reasoning models. These results suggest that the model has some generalization ability to long and complex videos, even under current data limitations.
>
> |               | VideoMME (long) | LongVideo-Reason-eval |
> | --------------- | ----------------- | ----------------------- |
> | Qwen2.5-VL-7B | 50.8            | 59.3                  |
> | Video-R1      | 50.7            | 68.9                  |
> | VideoRFT      | 50.6            | **69.4**        |
> | Open-o3 Video | **54.9**  | **69.4**        |
>
> *Table: Results on VideoMME-long and LongVideo-Reason-Eval.*

---

> ### Author Response · Authors · 2025-11-23
> **Official Comment by Authors (Part2)**
>
> **W3 and Q3: About performance on long videos.**
>
> As also stated in Appendix A.9 and explanation in W2, adapting the data and model to more complex scenes, such as very long videos, small objects, and rapid motion, is an important direction for our future research.
>
> At the same time, we have already conducted experiments on ​**LongVideo-Reason-eval**​, which contains many long videos (most are exceed 10 minutes, some are close to 30 minutes, 1 hour). The model shows improvements over the baseline and achieves performance ​**close to models trained with much larger reasoning datasets**​, such as Video-R1 and VideoRFT. To balance performance and efficiency, we evaluate using both 64 frames and 16 frames. Even with only 16 frames, Open-o3 Video can better utilize limited information and performs better than Video-R1 and VideoRFT.
>
> |               | Frames | LongVideo-Reason-eval |
> | --------------- | -------- | ----------------------- |
> | Qwen2.5-VL    | 64     | 59.3%                 |
> | Video-R1      | 64/16  | 68.9%/67.3%           |
> | VideoRFT      | 64/16  | 69.4%/68.0%           |
> | Open-o3 Video | 64/16  | 69.4%/69.2%           |
>
> *Table: Results on LongVideo-Reason-Eval under different frame numbers and key-frame selection settings.*
>
>
>
> **Q2: About adding audio information.**
>
> Due to the limitations of the base model (Qwen2.5-VL-7B), our model does not support direct audio input. And aligning temporal cues, spatial localization, audio content, and reasoning text is also a key part of our *Limitations and Future Work* discussion (Appendix A.9).
>
>
> However, ASR can be viewed as an alternative form of audio information. When we evaluate ​**VideoMME with subtitles**​, the model achieves ​70.9%​, which is +7.3% compared to the setting without subtitles, and +1.6% higher than the Qwen2.5-VL baseline. This suggests that ASR information provides useful additional signals for evidence-reasoning.
>
> |                | **Overall ​(w/o sub)** | **Overall (w/sub)** |
> | --------------- | --------------------- | ------------------------- |
> | Qwen2.5-VL    | 62.4            | 69.3                    |
> | Open-o3 Video | 63.6            | 70.9                    |
>
> *Table: Performance on VideoMME. “w/sub” denotes using ​*​*ASR*​*​​ subtitles; “​*​*w/o*​*​ sub” uses video-only inputs. ​*
>
>
> In summary, we thank the reviewer again for the constructive comments. We have clarified the limitations of our current data, discussed the model’s behavior on long and complex videos, and explained the potential benefits of audio information. These discussions are reflected in the revised version of the paper. We hope that our responses help address the concerns raised by the reviewer.
>
> **References:**
>
> [1] CameraBench: Towards Understanding Camera Motions in Any Video, NeurIPS 2025
>
> [2] Scaling RL to Long Videos, NeurIPS 2025

---

### Official Review · Reviewer_kwb2 · 2025-10-29

**Soundness:** 3
**Presentation:** 2
**Contribution:** 2
**Rating:** 4
**Confidence:** 3

**Summary:**

The paper describes a new, automatically generated dataset and training recipe for fine-tuning a vision-language model on video reasoning tasks. It also introduces a new model, based on fine-tuning Qwen2.5-VL-7B using this data and recipe. The goal of the data is to improve spatio-temporal grounding and the benefits of the approach are shown on various benchmarks (V-STAR, VideoMME, WorldSense, and others).

**Strengths:**

The paper shows results based on labor-intensive data engineering and training efforts, and the results on the selected benchmarks are strong. Reasoning over videos utilizing spatial, temporal and/or spatio-temporal grounding are difficult open problems.

**Weaknesses:**

The kind of automated data curation described in the paper seems more of an engineering effort to increase benchmark performance than an insightful scientific investigation of AI models, their limitations, etc. The data and reward generation appear quite finnicky with many hyperparameters, etc. While this not a problem per se, in line with the previous comment, it seems to come at the cost of scientific depth or insights.

**Questions:**

What exactly is meant by “Self-consistency Checking” (line 242), that is, how is it performed?

(line 256) “We initialize our framework from Qwen2.5-VL-7B” seems a bit strange and overblown. Do you mean “initialize our model”?

I would like to question the statement that the introduced model’s outperforming frontier models, like GPT-4o and Gemini, shows “significant advances in temporal and spatial grounding”. Isn’t this simply a matter of having been trained on this type of data/task? This is also shown by the fact that even the base model (Qwen2.5-VL-7B) outperforms the frontier models in some of the relevant metrics.

How were the models from Table 2 chosen in comparison to Table 1. Are models from Table 1 (like InternVL or Video-Llama3) not applicable here?

How does this work compare to earlier (and simpler) work on spatio-temporal grounding based on supervised learning (for example, “Look, Remember and Reason: Grounded reasoning in videos with language models”, Bhattacharyya et al. 2024, and similar work)?

The paper uses a lot of boasting language, which I find makes it harder to follow than necessary (“We have meticulously curated”,  “Through this powerful combination of curated data and tailored training”, “our approach brings significant advances in temporal and spatial grounding”, etc.). In my mind, the paper would be stronger and easier to read if it simply stated facts.

How were hyperparameters like sigma, reward weightings, data mixes, etc., determined?

---

> ### Author Response · Authors · 2025-11-23
> **Official Comment by Authors (Part1)**
>
> We sincerely thank the reviewer for the detailed and constructive comments. We appreciate the recognition that our work addresses a difficult and important open problem in video reasoning. We respond to each weakness and question below.
>
> **W1: About scientific depth and engineering effort.**
>
> We thank the reviewer for this concern. We clarify the scientific depth and insight of our work from two perspectives:
>
> **(1) Achieving spatio-temporal grounded reasoning is challenging itself.** Most existing video reasoning models rely on text-only chains of thought. In contrast, enabling a model to produce timestamps and bounding boxes within the reasoning trace is non-trivial:
>
> * Current datasets that provide synchronized spatio-temporal grounded reasoning annotations are scarce. Constructing such annotations is necessary to study how reasoning text aligns with temporal and spatial cues.
> * RL rewards for temporal and spatial alignment are sparse and unstable unless carefully designed. Moreover, the hyperparameters used in our RL stage are essential to training stability. We have conducted ablations on important hyperparameters in the paper.
>
> Therefore, achieving this structured reasoning format not only requires engineering effort, but also demands technical innovations in data construction, and reward design.
>
> **(2) The grounded reasoning format addresses a scientific limitation in video understanding.** Purely textual reasoning does not reveal **when** or **where** the model obtains visual evidence, making it difficult to determine whether the model actually finds key information or hallucinates irrelevant content. **Such textual traces are not intuitive  enough and hard to verify.** In contrast, our grounded reasoning format integrates timestamps and bounding boxes into the chain of thought, it brings several advantages:
>
> * Helpful to improve interpretability, verifiability, and reliability, and can support test-time scaling by evidence-based quality assessment (Appendix A.7).
> * Enables more accurate answers by locating and using the correct visual cues. It improves perception and spatial-temporal grounding while maintaining basic QA capability.
>
> In summary, the proposed **spatio-temporal grounded reasoning ​**is a meaningful problem. It directly targets the limitations of current video CoT, extends the "thinking with imagee" paradigm to more complex video scenarios, and tries to tackle challenges in accurate spatio-temporal localization and "visual evidence-reasoning text" alignment.
>
> **Q1: About “Self-consistency Checking”**
>
> We thank the reviewer for asking for clarification. Self-consistency checking includes the following:
>
> - Our annotations contain timestamps, bounding boxes, and a spatio-temporal reasoning chain. We ensure that all boxes and timestamps mentioned in the reasoning chain appear in the “key\_frames” and “key\_items” fields. If some annotations are missing, we either remove or supplement them to ensure every referenced entity has temporal and spatial grounding.
>
> - We retrieve the frames and boxes mentioned in the reasoning chain, crop them, and check whether they are relevant to the corresponding reasoning text sentence. If they are not aligned, the sample is removed. This ensures consistency between temporal annotations, spatial annotations, and reasoning text.
>
> We add detailed explanation of this step in the revised paper.
>
> **Q2: About “initialize our framework”**
>
> We thank the reviewer for pointing this out. We agree that the phrase is not appropriate. We have corrected it to **“initialize our model” ​**in the revised paper.

---

> ### Author Response · Authors · 2025-11-23
> **Official Comment by Authors (Part2)**
>
> **Q3: About the claim "outperforming frontier models" and "significant advances in temporal and ​spatial grounding"**
>
> We thank the reviewer for the critical comment. V-STAR is designed to reveal weaknesses in frontier proprietary models and current open-source video base models, especially regarding temporal and spatial grounding through “what”, “when”, and “where” tasks. The results presented in the original V-STAR paper reflect that this is an important and underexplored problem in the field.
>
> Our work is not intended to build a general video foundation model like Qwen-VL series or InternVL series. Our goal is to preserve the base model’s QA ability while adding and improving temporal and spatial grounding. Tables 1 and 2 in the paper show that our model better solves “when” and “where” tasks compared with the base model, and that it maintains or even improves performance on general video understanding benchmarks such as VideoMME and VideoMMMU.
>
> We agree that the performance gains depend on training with grounding data and spatio-temporal reasoning annotations. However, experiments demonstrate that our method can add grounding capabilities without harming general QA ability, and that explicit evidence-based reasoning is useful.
>
> What's more, we apologize for the statements (too strong) in the paper. We have revised the sentence to a more factual form in the revised paper, as shown below:
>
> *"Overall, compared with the Qwen2.5-VL baseline, our model improves performance by +14.4% mAM and +24.2% mLGM on V-STAR. It further surpasses proprietary models such as GPT-4o and Gemini-2-Flash and achieves state-of-the-art performance."*
>
> **Q4: About model choices in Table 2.**
>
> In Table 2, we compare our model with **Video-R1** [1] and ​**VideoRFT** [2], which are representative video reasoning models this year. They use the SFT+RL paradigm for text-only reasoning. Since Open-o3 Video is also a video reasoning model, these comparisons are appropriate. Models such as InternVL and VideoLLaMA3 are video foundation models, similar to our baseline Qwen2.5-VL. Because our goal is to demonstrate the effectiveness of adding spatio-temporal reasoning capabilities based on the video foundation model for QA tasks, we mainly compare with the base model itself and with video reasoning models that are closely related to our approach.
>
>
>
> **Q5: Comparison to prior supervised spatio-temporal grounding work**
>
> Open-o3 Video and earlier work such as LRR [3] (the reviewer mentioned) share the goal of improving temporal and spatial grounding. There are, however, several differences:
>
> **(1)Task focus: ​** LRR focuses on object detection, re-identification, and tracking to improve action recognition and visual causal reasoning. Open-o3 Video supports complex QA tasks and integrates localization into the reasoning process.
>
> **(2)Output form: ​** LRR outputs task-specific predictions. Open-o3 Video generates a grounded reasoning chain + final answer.
>
> **(3)Training method: ​** LRR modifies the architecture with a two-stream encoder and top-down cross-attention, trained in a supervised manner. Open-o3 Video keeps the Qwen2.5-VL architecture unchanged and performs post-training with spatio-temporal reasoning data using SFT+RL.
>
> **(4)Model scale: ​** LRR is much smaller. Open-o3 Video is based on a 7B multimodal base model and aims for broader applicability.
>
> Earlier work such as **STCAT [4]** also performs spatio-temporal grounding via modifying the transformer architecture, but without multimodal foundation model support, it cannot generalize to diverse VideoQA tasks.
>
> Despite the emergence of powerful MLLMs in recent two years, earlier work such as LRR still provides important inspiration. These studies remind us that, while focusing on semantic understanding, we should also deepen the investigation of temporal and spatial grounding. They highlight that “look” and “remember” are essential foundations for achieving “reason.”
>
> We have added these earlier spatio-temporal grounding works (e.g., STCAT and LRR) to the **Related Works​** section of the revised paper.

---

> > ### Comment · Reviewer_kwb2 · 2025-11-27
> >
> > Thank you for the detailed clarifications.
> >
> > I still find it somewhat strange to selectively drop models like InternVL or Video-Llama3 from the comparison in Table 2 by arguing that they are less relevant. Why not include the results and let the reader decide what to make of the results?

---

> > > ### Author Response · Authors · 2025-12-01
> > >
> > > Thank you for your suggestion. We agree that adding more video understanding models can provide a clearer comparison for readers.
> > >
> > > We have now included the results of **InternVL-2.5-8B** and **VideoLLaMA3-7B** in Table 2, using the same experimental settings as other models. These results show that VideoLLaMA-3 performs better on temporal grounding, while InternVL2.5 is stronger on WorldSense.  Across all general video understanding and temporal grounding benchmarks, our model achieves relatively higher average scores among models of similar scale.
> > >
> > > We hope this update can help to address your concern. Thank you!
> > >
> > > | Model                | VideoMME Overall | VideoMME Long | WorldSense Overall | WorldSense Recognition | VideoMMMU Overall | VideoMMMU Perception | LRR Acc | TVGBench mIoU | Avg  |
> > > |----------------------|------------------|----------------|---------------------|-------------------------|--------------------|------------------------|---------|----------------|------|
> > > | GPT-4o               | 71.9             | -              | 42.6                | -                       | 61.2               | 66.0                   | -       | -              | -    |
> > > | VideoLLaMA3-7B       | 60.6             | 48.7           | 37.3                | 38.1                    | 46.5               | 59.7                   | 59.8    | **22.2**       | 45.3 |
> > > | InternVL-2.5-8B      | 62.3             | 51.2           | **39.6**            | **38.5**                | 42.4               | 57.0                   | 62.0    | 6.3            | 42.5 |
> > > | **Qwen2.5-VL-7B (base)** | 62.4         | 50.8           | 36.1                | 33.7                    | 51.2               | 64.7                   | 59.3    | 16.3           | 45.1 |
> > > | VideoRFT-7B          | 59.8             | 50.7           | 38.2                | 36.6                    | 51.1               | 66.0                   | **69.4**| 14.3           | 46.6 |
> > > | VideoR1-7B           | 61.4             | 50.6           | 35.5                | 32.8                    | 52.4               | 65.3                   | 68.9    | 9.6            | 45.6 |
> > > | **Open-o3 Video (Ours)** | **63.6**      | **54.9**       | 37.5                | 36.8                    | **52.3**           | **68.0**               | **69.4**| 20.8       | **48.7** |
> > > | v.s. Qwen2.5-VL-7B   | ↑ 1.2            | ↑ 4.1          | ↑ 1.4               | ↑ 3.1                   | ↑ 1.1              | ↑ 3.3                  | ↑ 10.1  | ↑ 4.5          | ↑ 3.6 |
> > >
> > > *Table2: Performance across different video understanding and temporal grounding benchmarks.*

---

> ### Author Response · Authors · 2025-11-23
> **Official Comment by Authors (Part3)**
>
> **Q6: About "boasting language"**
>
> We sincerely apologize for this issue. **We have revised the writing to reduce unnecessary adjectives and adverbs.** For example, the following sentences the reviewer mentioned have been corrected:
>
> **(1) ​**We have curated two datasets, STGR-CoT-30k and STGR-RL-36k, for supervised fine-tuning and reinforcement learning, respectively.
>
> **(2) ​**Through this combination of curated data and our training procedure, Open-o3 Video produces reasoning that is accurate, interpretable, and grounded in the visual evidence.
>
> **(3) ​**Overall, compared with the Qwen2.5-VL baseline, our model improves performance by +14.4% mAM and +24.2% mLGM on V-STAR. It further surpasses proprietary models such as GPT-4o and Gemini-2-Flash and achieves state-of-the-art performance.
>
>
>
> **Q7: About the ablation and explanation of hyperparameters**
>
> 1. **Sigma (σ): ​**As described in the paper, we use an adaptive schedule: σ decreases from 4 to 1 and then stays at 1.  In the reward function, y = exp(-(x²)/(2σ²)), x is the difference between predicted and ground-truth timestamps:
>    - If σ = 1, the reward becomes very small (<0.1) for errors within 3s (x=3), leading to sparse reward early in training.
>    - If σ = 4, timestamps receive high reward (>0.6) even within 4s error (x=4), receive >0.9 within 2s error (x=2), making it hard to further reduce error.
>       Therefore, the adaptive schedule helps the model move from coarse to fine localization. Ablations show that σ=1 and σ=4 both perform worse than the adaptive schedule.
>
> | V-STAR        | mAM            | mLGM           |
> | --------------- | ---------------- | ------------- |
> | Open-o3 Video | **33.7** | **46.6** |
> | sigma=1.0     | 32.6           | 44.5           |
> | sigma=4.0     | 33.0           | 45.2           |
>
> *Table: Ablation on the adaptive temporal proximity parameter σ, showing that both fixed σ=1 and σ=4 underperform the adaptive one used in Open-o3 Video.*
>
>
> 2. **Reward weightings: ​**Each reward term is naturally in the range [0,1], so we just use a simple sum without extra hyperparameters. When the model does not use o3-style grounding (i.e., no thinking reward), performance drops clearly on V-STAR, confirming the importance of grounding-based reward.
>
> |             | what(acc) | chain1-tiou | chain1-viou | chain2-tiou | chain2-viou | mAM   | mLGM  |
> | --------------- | ----------- | ------------- | ------------- | ------------- | ------------- | ------- | ------- |
> | Open-o3 Video | 61.03     | 24.53       | 25.37       | 23.97       | 5.97        | 33.65 | 46.58 |
> | Non-o3-like   | 58.60     | 23.08       | 13.74       | 22.70       | 3.09        | 29.97 | 41.04 |
>
> *Table: Performance between Open-o3 Video and a Non-o3-like variant. “Non-o3-like” refers to removing spatio-temporal grounded reasoning (no grounded ​SFT data and no thinking rewards in ​RL). It is a text-only reasoning model.*
>
>
> 3. **Data mixes: ​**We study data composition in Section 5.2 (Table 5) and Appendix A.2 (Table 6). Results show the importance of spatio-temporal reasoning data and the benefits of an appropriate ratio of VideoQA data.
>
> In summary, we thank the reviewer again for the careful reading and thoughtful feedback. We have revised the paper to clarify scientific motivation, expand the related works section, correct writing issues, and provide more detail about consistency checking and hyperparameters. We hope that these clarifications help address the reviewer’s concerns.
>
> **References:**
>
> [1] Video-R1: Reinforcing Video Reasoning in MLLMs, NeurIPS 2025
>
> [2] VideoRFT: Incentivizing Video Reasoning Capability in MLLMs via Reinforced Fine-Tuning, NeurIPS 2025
>
> [3] Look, Remember and Reason: Grounded Reasoning in Videos with Language Models, ICLR 2024
>
> [4] Embracing Consistency: A One-Stage Approach for Spatio-Temporal Video Grounding, NeurIPS 2022

---

### Official Review · Reviewer_PPw7 · 2025-10-30

**Soundness:** 3
**Presentation:** 3
**Contribution:** 2
**Rating:** 6
**Confidence:** 4

**Summary:**

This paper introduces Open-o3 Video, a framework designed to enhance video reasoning in large multimodal models by grounding textual answers in explicit spatio-temporal evidence. Addressing the limitations of prior models that only produce text-based rationales, Open-o3 Video generates answers alongside key timestamps and bounding boxes that pinpoint the visual evidence supporting its conclusions. The authors identify two primary challenges: the absence of high-quality datasets with joint spatio-temporal supervision and the difficulty of training a model to perform simultaneous temporal tracking and spatial localization. Empirically, Open-o3 Video, built upon the Qwen2.5-VL-7B model, achieves state-of-the-art results on the V-STAR benchmark, significantly outperforming its base model by +14.4% mAM and +24.2% mLGM, and also surpassing proprietary models like GPT-4o. The model also shows consistent improvements on other video understanding benchmarks such as VideoMME, WorldSense, and TVGBench.

**Strengths:**

1. The paper addresses a critical and timely problem in multimodal AI: moving beyond opaque, text-only reasoning to verifiable, evidence-grounded reasoning. Extending the "thinking with images" paradigm to the video domain is a non-trivial and important research direction. The proposed reward mechanisms—adaptive temporal proximity and temporal gating—are novel, well-motivated, and directly target core challenges in learning spatio-temporal grounding.

2. The paper is methodologically sound. The two-stage training approach is well-justified, and the choice of GSPO for sequence-level optimization is appropriate for the complex, long-form output. The ablation studies in Section 5.2 are particularly strong; they clearly isolate and validate the contribution of each component: the SFT+RL strategy, the specific reward designs (Ada. and Gat.), and the high-quality annotated data.

3. The creation and planned release of the STGR-CoT-30k and STGR-RL-36k datasets represent a major contribution to the research community. The detailed description of the data annotation pipeline (Figure 2, Appendix A.3) provides transparency.

**Weaknesses:**

1. The model processes videos by sampling 16 frames uniformly, with the addition of key frames during training. This is a sparse representation that may not be sufficient for very long videos or for scenarios where critical evidence is extremely brief and could be missed by the sampling. While the authors acknowledge this limitation, its practical impact on more complex, real-world videos remains an open question.

2. The model is trained to generate a highly specific output format (<obj>...</obj><box>...</box>at<t>...</t>s). This type of instruction-following can sometimes be brittle. The paper does not discuss how the model behaves if it fails to adhere to this format or how such failures are handled during evaluation and RL training

3. Seems the authors missed some existing video reasoning benchmarks, e.g. LongVideo-Reason-eval [1]

[1] Scaling RL to Long Videos. Neurips 2025. https://huggingface.co/datasets/LongVideo-Reason/longvideo_eval_videos

**Questions:**

1. The strategy of uniformly sampling 16 frames seems sparse for longer videos. Could you elaborate on how adding annotated key frames during training helps the model generalize to finding unseen and potentially brief key moments at test time? Have you experimented with alternative, more dynamic frame selection strategies at inference?

---

> ### Author Response · Authors · 2025-11-23
> **Official Comment by Authors**
>
> We sincerely thank the reviewer PPw7 for the detailed and constructive feedback. We are encouraged by the reviewer’s recognition that our paper tackles a critical and non-trivial problem ("Thinking with videos") in multimodal AI.  We are also grateful for the acknowledgement that our reward mechanisms are novel and well motivated. Below, we provide responses to the weaknesses and questions.
>
> **W1 and Q1: About the frame sampling and generalization.**
>
> Our use of 16 uniform frames together with annotated key frames can be explained from two perspectives:
>
> 1. Since most videos in our training set are shorter than 3 minutes and the training efficiency,  we do not conduct denser sampling in the training process. We acknowledge that the current training data is relatively short, and there is clear room for improvement on longer and more complex videos. In future work, we plan to construct annotated datasets with longer videos and adopt higher frame rates to further improve performance. However, although the model is not trained on long videos, it still shows performance advantages on **VideoMME-Long** and ​**LongVideo-Reason-eval** [1]​, improving over the Qwen2.5-VL baseline by +4.1% and ​+10.1%  ​(Table 2 in the revised paper and in the response of W3)​, respectively. Even when using only 16 frames at inference, our model surpasses Video-R1 and VideoRFT on LongVideo-Reason-eval by 1.9% and ​1.2%​.
>
> 2. The CoT annotations in our training data contain grounded reasoning of the form `<obj>...</obj><box>...</box>at<t>timestamp</t>s`, which includes key-frame information. The visual content of these key frames should be provided to the model. Otherwise, it cannot align the textual grounding format with the actual temporal and spatial content, and it will be difficult for the model to learn precise localization. At inference time, we acknowledge that uniform sampling cannot guarantee capturing key evidence. It is a simple strategy, but it can still work well. In addition, we also experiment with **keyframe sampling** at inference. Following AKS [2] , we select key frames during inference. This yields ​70.1%, an improvement of +0.7% over uniform sampling.
>
>
> **W2: About the grounding format and its robustness.**
>
> We define a unified grounding format: `<obj>object_name</obj><box>bounding_box</box>at<t>timestamp</t>s`. This format can be primarily learned during the cold-start SFT stage. This helps RL training, because the model can obtain rewards related to temporal and spatial grounding more quickly. During RL, we include a format reward. Only responses that strictly follow this structure receive the higher reward. After RL training, the format reward stabilizes above 0.99, showing that the model follows this format in almost all cases.
>
> At inference, there may still be occasional format errors. Our test-time scaling procedure (Appendix A.7) addresses this. We map the predicted grounding to the actual video frames, crop the regions, and ask the model whether the evidence is relevant to the question. If the evidence is irrelevant or if no evidence is produced, we mark that reasoning path as low-quality and assign it a smaller weight during voting. Experiments show that this weighted voting performs better than naive majority voting (Table 9).
>
>
> **W3: About the LongVideo-Reason-eval benchmark.**
>
> We conduct experiments on **LongVideo-Reason-eval. ​**Our model improves over the Qwen baseline by +10.1%, matches the performance of Video-R1 and VideoRFT at 64 frames, and performs better when fewer frames (16) are used. Even our method is not designed form long-video setting, our method can still achieve better results than Video-R1 and VideoRFT.
>
> |              | Frames | LongVideo-Reason-eval |
> | --------------- | -------- | ----------------------- |
> | Qwen2.5-VL    | 64     | 59.3%                 |
> | Video-R1      | 64/16  | 68.9%/67.3%           |
> | VideoRFT      | 64/16  | 69.4%/68.0%           |
> | Open-o3 Video | 64/16  | 69.4%/69.2%           |
>
> *Table: Results on LongVideo-Reason-Eval under different frame numbers.*
>
> In summary, we thank the reviewer again for the valuable comments. We expand our explanations of the sampling strategy, clarify the role of key frames, discuss format robustness in training and inference, and include detailed results on video reasoning benchmark LongVideo-Reason-eval. We hope that these clarifications help address the concerns raised by the reviewer.
>
> **References:**
>
> [1] Scaling RL to Long Videos, NeurIPS 2025
>
> [2] Adaptive Keyframe Sampling for Long Video Understanding, CVPR 2025

---

### Official Review · Reviewer_xbZp · 2025-11-01

**Soundness:** 3
**Presentation:** 3
**Contribution:** 2
**Rating:** 6
**Confidence:** 4

**Summary:**

The paper aims to introduce a unified framework for grounded video reasoning that integrates explicit spatio-temporal evidence through timestamped frames and localized bounding boxes. The paper proposed the STGR-CoT-30k, STGR-RL-36k datasets and uses a with a two-stage training strategy using supervised fine-tuning and GSPO-based reinforcement learning. Experiments on V-STAR show promising performance.

**Strengths:**

* The proposed STGR-CoT-30k and STGR-RL-36k datasets for supervised fine-tuning and RL, are novel and interesting, would add value to the community.
* The proposed approach leads to improved performance on the V-STAR benchmark.
* The paper is well written and easy to understand.

**Weaknesses:**

• Novelty: The dataset generation and training scheme – supervised fine-tuning followed by GRPO based RL training is largely based on prior work, e.g., the Deepseek family of models and more recently by papers such as "Scaling RL to Long Videos, NeurIPS 2025". The novelty of the training scheme should be discussed in more detail.

* Performance: The proposed approach leads to limited gains on the VideoMME or VideoMMMU benchmarks (~1%) vs the base Qwen model. Moreover, state of the art models such VideoLLAMA3 obtains 66.2 % accuracy.

* Limited evaluation datasets: The paper only evaluates on the V-STAR benchmark. For fair comparison to prior work the paper should include evaluation on additional benchmarks that focus specially on grounding such as "STAR: A Benchmark for Situated Reasoning in Real-World Videos, NeurIPS 2021".

* The paper should discuss prior work on grounding to fine-grained visual information in videos such as: "Look, Remember and Reason: Grounded reasoning in videos with language models, ICLR 2024"; "Fine-grained Spatiotemporal Grounding on Egocentric Videos, ICCV 2025".

• The model seems to rely on a stable and low frame rate, it is unclear if the model can deal with variable or high frame rates.

**Questions:**

* The performance on benchmarks such as VideoMME or VideoMMMU should be discussed in more detail.
* The paper should add additional evaluation on benchmarks such as STAR.
* The paper should include a more extensive discussion on prior works on video grounding.

---

> ### Author Response · Authors · 2025-11-23
> **Official Comment by Authors (Part 1)**
>
> We sincerely thank the reviewer xbZp for the constructive and detailed feedback. We truly appreciate the positive assessment of our work, especially the recognition that our proposed datasets “would add value to the community.” This is highly encouraging, as we are committed to further advancing and expanding research in this direction. We are also grateful to the reviewer for pointing out the weaknesses of our current work. Below, we provide detailed clarifications and additional explanations for each of the concerns raised.
>
> **W1: About the novelty issue.**
>
> We agree with the reviewer that many video reasoning works in the year of 2025 adopt the SFT + RL training paradigm, such as LongVILA (mentioned by the reviewer), as well as Video-R1 and VideoRFT (included in our comparisons). However, the training paradigm is not the novelty of our work.
>
> Our key technical contribution is to enable a model to **produce explicit spatio-temporal evidence during the reasoning process. ​**It can allow**​ ​**the reasoning to be more transparent, intuitive, verifiable, and supportive of test-time scaling.  **To the best of our knowledge, we are the first to achieve this.**
>
> To realize this capability, our work introduces the following innovations:
>
> 1. **A new data construction pipeline** that generates unified spatio-temporal reasoning data. The data is carefully designed with Gemini 2.5 Pro annotation, filtering and checking, and it covers timestamps, bounding boxes, and grounded reasoning traces.
> 2. **A new reward ​designed​ specifically for joint temporal and ​spatial grounding. ​**It includes ​**adaptive temporal proximity**​, which stabilizes early training and gradually encourages precise alignment, and ​**temporal gating**​, which ensures spatial rewards are only applied when temporal predictions are reliable. They are essential for tasks that require simultaneous temporal and spatial localization.
>
> We also find that if we remove this “o3-style” reasoning (with no spatio-temporal evidence provided in SFT, and no evidence-related rewards in RL),  the model’s performance decreases (Table below). It shows that our proposed spatio-temporal reasoning capability offers advantages over text-only reasoning. It also confirms the effectiveness of both our constructed spatio-temporal data and the grounding-based thinking reward design.
>
> |             | V-STAR(what) | V-STAR(mLGM) | VideoMME | VideoMME(long) | WorldSense | WorldSense (Recognition) |
> | --------------- | -------------- | -------------- | ----------- | ---------------- | ------------- | -------------------------- |
> | Open-o3 Video | 61.0         | 46.6         | 63.6      | 54.9           | 37.5        | 36.8                     |
> | Non-o3-like   | 58.6         | 41.0         | 62.3      | 52.8           | 37.1        | 36.4                     |
>
> *Table: Performance  between Open-o3 Video and a Non-o3-like variant. “Non-o3-like” refers to removing spatio-temporal grounded reasoning (no grounded ​SFT data and no thinking rewards in ​RL). It is a text-only reasoning model.*

---

> ### Author Response · Authors · 2025-11-23
> **Official Comment by Authors (Part 2)**
>
> **W2 and Q1:  About the performance issue on video understanding benchmarks.**
>
> The objective of our work is to improve ​**spatio-temporal grounding ability**​, while at the same time ​**maintaining or improving basic VideoQA performance**​.
>
> For general VideoQA tasks, unlike large video–language foundation models trained with extensive QA data (e.g., the InternVL series, Qwen-VL series, and VideoLLaMA series), we acknowledge that our model cannot achieve 7B-level SOTA performance on most video understanding benchmarks.
>
> Compared with Video-R1, VideoRFT, and VideoLLaMA3, which finetune on a much larger VideoQA dataset (260k, 310k, and over 1M samples, respectively), our model uses only 15k VideoQA samples. Even with less QA supervision, Open-o3 Video can enhance localization ability, provide temporal and spatial evidence, and still maintain (somtimes surpass) the baseline model’s QA ability.
>
> A more detailed result also supports our model's performance. Our model shows gains on perception-oriented subsets since it focuses more on grounding. With 64 sampled frames, it improves **VideoMMMU–Perception** by ​+3.3%​, and improves **WorldSense–Recognition** by ​+3.1%. (Table 2 in paper)
>
> As for the reviewer’s comment on VideoLLaMA3-7B (achieving 66.2% on VideoMME (w/o sub)) , our model based on Qwen2.5-VL-7B does not surpass VideoLLaMA3 under the w/o sub setting. However, our experiments show that when ASR information is included, our model achieves 70.9%  (w/ sub setting), which is comparable to VideoLLaMA3 (70.3%).
>
> |                   | Overall w/o sub | Overall w/sub |
> | ----------------------- | ----------------- | --------------- |
> | Qwen2.5-VL (baseline) | 62.4            | 69.3          |
> | VideoLLaMA3           | 66.2            | 70.3          |
> | Open-o3 Video         | 63.6            | 70.9          |
>
> *Table: Performance on VideoMME. “w/sub” denotes using ​*​*ASR*​*​​ subtitles; “​*​*w/o*​*​ sub” uses video-only inputs. Open-o3 Video is comparable to VideoLLaMA3 when ASR information is included.*
>
>
>
> **W3 and Q2: About the evaluation on grounding benchmark STAR.**
>
> We thank the reviewer for the suggestion about evaluation of additional grounding-focused benchmarks. STAR**[1]** is indeed an important benchmark for situated reasoning in real-world videos, and we agree that including it can provide a more complete view of the model’s grounding ability.
>
> Existing models such as LRR usually rely on specialized training procedures, including object-recognition surrogate tasks, to adapt to STAR’s requirements. In contrast, our method does not use STAR training data or any additional adaptation. We evaluate the model in a **zero-shot** manner on the STAR validation set (the test set does not provide labels). The results are as follows:
>
> * Qwen2.5-VL-7B (base): 67.27%
> * Open-o3 Video: 70.54%
>
> This shows that our method improves over the baseline model by ​3.3%, and its performance is close to specialized models.

---

> ### Author Response · Authors · 2025-11-23
> **Official Comment by Authors (Part 3)**
>
> **W4 and Q3:  About related works on video spatio-temporal grounding.**
>
> We thank the reviewer for pointing this out. Early works such as **STCAT**[2] and **LRR**[3] also improved spatio-temporal grounding by optimizing transformer-based multimodal architectures. **LRR** uses a two-stream encoder together with an LM backbone equipped with top-down cross-attention, enabling a three-stage “look–remember–reason” process. Recent methods like ​**EgoMask**[4]​, built on models such as Sa2Va and VideoLISA, enhanced grounding for ego-centric videos through fine-tuning. However, these approaches typically output only detection boxes or fixed-format predictions. They cannot be broadly applied to diverse VideoQA tasks, nor do they combine grounding ability with a chain-of-thought reasoning process as our method does.
>
> We have updated the **Related Works** section of the paper to incorporate a more complete discussion and citations of these prior works.
>
>
>
> **W5:  About the issue of the frame rate.**
>
> We thank the reviewer for raising this concern.
>
> 1.  On the question of high versus low frame rates, we conduct experiments on the **LongVideo-Reason-Eval​**[5] benchmark, which contains long, information-dense, and visually complex videos. As shown in the table below, we find that higher frame rates can bring some improvements. However, even with only **16 ​frames**​, our method already achieves strong performance and surpasses the baseline (Qwen2.5VL) and other reasoning models (Video-R1, VideoRFT). Moreover, increasing the frame rate further leads to only limited marginal gains.
>
> 2. About ​**variable frame rates**​, following the prior work ​**AKS**​[6], we apply a keyframe selection strategy. We retain the **64 most informative ​**​**frames** for answering the question, which helps the model focus on the most relevant temporal and spatial cues. This setting yields an accuracy of ​70.1%, showing that variable frame rate selection (key frames) can provide marginal improvements. In addition, since our model supports multi-image reasoning with timestamps at inference, the model can still get the exact time of each selected frame even when the frame rate is not fixed.
>
>  |                  | Frames | LongVideo-Reason-eval |
> | -------------------- | -------- | ----------------------- |
> | Qwen2.5-VL         | 64     | 59.3%                 |
> | Video-R1           | 64/16  | 68.9%/67.3%           |
> | VideoRFT           | 64/16  | 69.4%/68.0%           |
> | Open-o3 Video      | 64/16  | 69.4%/69.2%           |
> | Open-o3 Video +AKS | 64     | 70.1%                 |
>
>
>
> *Table: Results on LongVideo-Reason-Eval under different frame numbers and key-frame selection settings.*
>
> In summary, the reviewer's suggestions help us clarify the novelty of this work. In the revised version, we add some additional evaluations (e.g., STAR, LongVideo-Reason-eval), expand the discussion of related work, and include more detailed analysis of model behavior under different frame-rate conditions. We hope that these clarifications can help address the questions raised by the reviewer.
>
> **References:**
>
> [1] STAR: A Benchmark for Situated Reasoning in Real-World Videos, NeurIPS 2021
>
> [2] Embracing Consistency: A One-Stage Approach for Spatio-Temporal Video Grounding, NeurIPS 2022
>
> [3] Look, Remember and Reason: Grounded Reasoning in Videos with Language Models, ICLR 2024
>
> [4] Fine-grained Spatiotemporal Grounding on Egocentric Videos, ICCV 2025
>
> [5] Scaling RL to Long Videos, NeurIPS 2025
>
> [6] Adaptive Keyframe Sampling for Long Video Understanding, CVPR 2025

---

### Author Response · Authors · 2025-11-23
**General Response and Paper Revision**

We sincerely thank all reviewers for their thoughtful and constructive feedback. We are especially encouraged by the recognition of our contributions. In particular, most reviewers highlighted the importance of our research problem, appreciated the value of our new datasets, and acknowledged our model’s effectiveness.



We also appreciate the reviewers’ helpful suggestions for improvement. We have tried our best effort to revise the paper in response to all concerns. All revisions are marked in red:

* **Related works:** We expand the discussion to include additional spatio-temporal grounding methods, such as STCAT, LRR, and EgoMask, and clarify the distinctions between our approach and prior works. (Section 2 in the paper)


* **Writing and clarity:** We improve the description of self-consistency checking (Section 3.2 in the paper) and revise redundant wording to improve readability. In particular, we have carried out line by line checking. (Section 1,4,5 in the paper) Please check our revised version.


* **Experimental evaluation:** In Table 2, we add results on a long video reasoning benchmark (LongVideo-Reason-Eval). And we also include additional video understanding models (VideoLLaMA3, Intern2.5-VL) for a more comprehensive comparison. In the Appendix (A.3, A.4) , we include evaluations on STAR and CameraBench, as well as ablation studies on frame rate. All newly added benchmark results are summarized below (more detailed results and additional experiments can be found in the revised paper and per-reviewer responses):



|  Models           | LongVideoReason-eval |     STAR    | Camerabench |
| --------------- | --------------------- | -------------| ------------- |
| Qwen2.5-VL-7B | 59.3                 | 67.3        | 57.5        |
| Open-o3 Video | 69.4 (+10.1)         | 70.5 (+3.2) | 58.8 (+1.3) |



These results show that Open-o3 Video has generalization ability in ​**long-video reasoning, grounding-focused situated reasoning, and different camera movements**​. In future work, we will continue to deepen this research to achieve better performance on long, complex and dynamic video scenarios.



In addition, we provide detailed responses to all reviewers' questions.

We thank the reviewers again for helping us strengthen the paper. We are always looking forward to open discussions. We will give our response as soon as possible once you raise more questions.


Sincerely,


Authors of Open-o3 Video

---

### Author Response · Authors · 2025-12-03
**Summary of the Rebuttal**

We sincerely thank the AC for the time and effort dedicated to reviewing our work. Below is a summary of our rebuttal:

1. Reviewers acknowledge the significance of our research problem, the proposed datasets, methods, and overall performance. We revise the paper to address all reviewers' concerns, and we provide a complete list of changes in the ***General Response and Paper Revision*** comment.
2. For the suggestions of **Reviewer xbZp: ​**We clarify the novelty of our research, compare Open-o3 Video with earlier video spatio-temporal grounding approaches, expand Table 2 with more video understanding models, and add STAR benchmark results and frame-rate ablation in the appendix.
3. For the suggestions of  **Reviewer PPw7: ​**We discuss the impact of frame rate, the robustness of our reasoning format, and the model’s generalization to long videos.
4. For the suggestions of  **Reviewer kwb2: ​**We emphasize the scientific value of the proposed spatio-temporal grounded reasoning paradigm, refine multiple sections to improve clarity of the paper, and further compare the differences and connections between our method and prior supervised grounding work. And we also enhance model comparisons in Table 2, and add hyper-parameters ablations in the appendix.
5. For the suggestions of  **Reviewer BpJt: ​**We present results on CameraBench to show our adaptive temporal proximity and temporal gating can generalize to different types of camera motion. We also add experiments on long-video benchmarks and discuss how ASR enhances evidence-based reasoning.
6. Among all reviewers, only reviewer kwb2 responds during the rebuttal period. The reviewer acknowledges our responses and provides follow-up suggestions. And we further expand Table 2 with additional video understanding baselines, and provide more comprehensive results.

Overall, we believe our method and dataset can make a valuable contribution to the research of **non-agent-based "thinking with videos"**, and empirical results support the effectiveness of our spatio-temporal grounded reasoning model.

Sincerely,

Authors of Open-o3 Video

---

### Meta-Review · Area_Chair_oumw · 2026-01-07

**Summary:**

The research presents a system for video reasoning alongside new datasets and training methods which help achieve this objective. The reviewers consider the research problem significant while finding the experimental findings encouraging yet they express multiple concerns about the study's failure to introduce new concepts beyond current SFT+RL frameworks and its dependence on intricate system design decisions and insufficient examination of model performance and restrictions.

**Reviewer Concerns:**

Multiple reviewers expressed doubts about the experimental comparisons because they thought the studies lacked fullness and fairness and used inappropriate baseline selection and the evaluation results did not demonstrate sufficient stability across different video conditions. The authors present complete answers together with supplementary research but the rebuttal fails to address the issues regarding originality and scientific complexity and broad applicability so the recommendation stands at rejection.

**Reviewer Scores:**

N/A

---

### Decision · Program_Chairs · 2026-01-26

Reject